# SPRI: Aligning Large Language Models with Context-Situated Principles

**Hongli Zhan** [† 1]  **Muneeza Azmat** [2]  **Raya Horesh** [2]  **Junyi Jessy Li** [1]  **Mikhail Yurochkin** [2 3]

## Abstract

Aligning Large Language Models to integrate and reflect human values, especially for tasks that demand intricate human oversight, is arduous since it is resource-intensive and time-consuming to depend on human expertise for context-specific guidance. Prior work has utilized predefined sets of rules or *principles* to steer the behavior of models (Bai et al., 2022b; Sun et al., 2023). However, these principles tend to be generic, making it challenging to adapt them to each individual input query or context. In this work, we present SITUATED-**PRI**NCIPLES (SPRI), a framework requiring minimal or no human effort that is designed to automatically generate guiding principles in real-time for each input query and utilize them to align each response. We evaluate SPRI on three tasks, and show that 1) SPRI can derive principles in a complex domain-specific task that leads to on-par performance as expert-crafted ones; 2) SPRI-generated principles lead to instance-specific rubrics that outperform prior LLM-as-a-judge frameworks; 3) using SPRI to generate synthetic SFT data leads to substantial improvement on truthfulness. We release our code and model generations at `https://github.com/honglizhan/SPRI-public`.

## 1. Introduction

Large Language Models (LLMs) have showcased impressive performance across diverse applications (Achiam et al., 2024; Dubey et al., 2024; Yang et al., 2025; Jiang et al., 2024; Groeneveld et al., 2024). However, in more complex tasks, human-expert-crafted prompts are required to

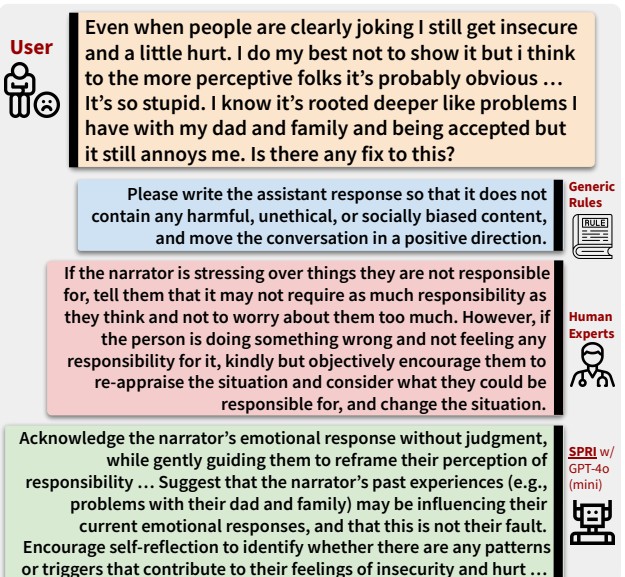

*Figure 1.* Using SPRI, `GPT-4o-mini` can generate situated and detailed principles to guide the response to a person narrating in distress. Compared with generic rules (Bai et al., 2022b) and human-expert-crafted principles (Zhan et al., 2024), SPRI requires minimal to no human efforts yet produces context-specific guidance for every query at hand.

achieve the desired level of performance. For example, Zhan et al. (2024) showed that LLMs are capable of generating high-quality cognitive reappraisals when guided by "constitutions" written by clinical psychologists with doctoral degrees.[1] LLM-as-a-judge (Zheng et al., 2023) is another prominent application that typically requires carefully crafted evaluation criteria to align with human annotators (Yu et al., 2024; Hashemi et al., 2024; Ye et al., 2024).

To better guide LLMs, several prior works utilized principles or constitutions in the context of synthetic data generation for alignment (Bai et al., 2022b; Sun et al., 2023). Such approaches are effective at reducing data annotation efforts, however, they are limited by the general nature of such principles making them hard to interpret in a given context, even for humans (Kirk et al., 2023a;b). For example, Bai

---

[†]Work started and partially done during Hongli's internship at IBM Research. [1]Department of Linguistics, The University of Texas at Austin, Austin, TX, USA [2]IBM Research, Yorktown Heights, NY, USA [3]MIT-IBM Watson AI Lab, Cambridge, MA, USA. Correspondence to: Hongli Zhan <honglizhan@utexas.edu>.

*Proceedings of the 42nd International Conference on Machine Learning*, Vancouver, Canada. PMLR 267, 2025. Copyright 2025 by the author(s).

---

[1]Cognitive reappraisal is a strategy commonly practiced by clinical psychologists to foster long-term emotional well-being (Arnold, 1960; Gross & John, 2003; Yeo & Ong, 2024). See Appendix §E for more details.

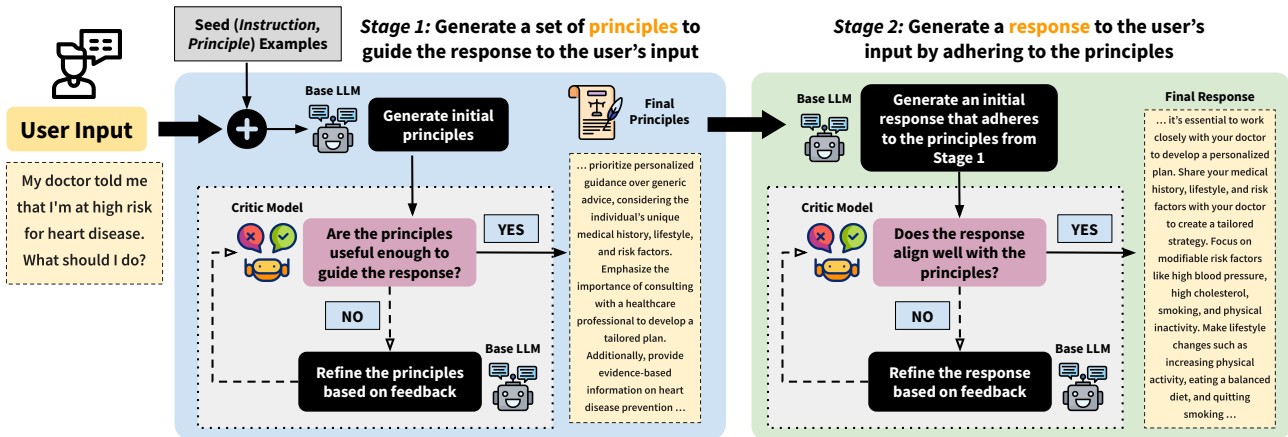

*Figure 2.* Overview for SPRI, which consists of two stages: 1) producing a set of principles specifically tailored to the user's input $T$, and 2) utilizing the generated principles to guide the response to $T$. Both stages include a critique-refine process involving a separate critic model, which aims to scrutinize the fitness of the principles to $T$ and the final responses' adherence to the generated principles.

et al. (2022b) employed the constitutional principle *"Identify specific ways in which the assistant's last response is harmful, unethical, racist, sexist, toxic, dangerous, or illegal"* to critique and refine model responses. The precise meaning of *harmful* or *unethical* is often situation-dependent limiting the effectiveness of the principle when aligning to nuanced human values. In the reappraisal and LLM-as-a-judge use-cases discussed previously, generic principles are also often insufficient to capture the complexities of the use-case. For example, Kim et al. (2025) use human annotators to craft instance-specific evaluation criteria for LLM judges for their open-ended generation benchmark, which is a considerable amount of human effort. We provide an example in the context of reappraisal in Figure 1.

We propose SITUATED-**PRI**NCIPLES (SPRI), a framework designed to automatically generate constitutional principles *specifically tailored to that input query* in real-time and utilize them to align each response. SPRI utilizes a base model and a critic model, and its algorithm consists of two stages. The first stage consists of a base model that comes up with principles and a critic model that helps the base model to iteratively refine the principles. The second stage then applies the principles to direct the base model's response to the specific user's input. The critic model reviews the response using the principles as criteria, and the base model adjusts the response according to the feedback from the critic model. Importantly, the critic model does *not* need to be stronger or larger than the base model. We illustrate our framework in Figure 2.

We evaluate SPRI in three situations:

(1) We consider a domain-specific task where expert-level complex principles were shown to be necessary: having LLMs produce cognitive reappraisals (§4.1). We show that models using principles derived from SPRI

perform on-par with those using principles crafted by professional psychologists.

(2) Evaluation of open-ended generations across complex tasks with LLM judges. We show that principles from SPRI result in correlation with human judgments on par with instance-specific human curated evaluation rubrics and outperform prior LLM-judge frameworks (§4.2).

(3) Generating synthetic data with SPRI proves effective for fine-tuning base LLMs, resulting in substantial improvement on TruthfulQA (Lin et al., 2022), whilst maintaining performance on other benchmarks (§5).

## 2. Related Work

**Scalable Oversight.** In order to minimize the amount of human oversight necessary to align LLMs, Bai et al. (2022b) introduced Constitutional AI, a method relying on a list of predefined hand-crafted rules or *constitutional principles* that aim to promote safe, reliable, and effective systems. Leveraging Reinforcement Learning from AI Feedback (RLAIF) (Lee et al., 2024), Constitutional AI uses these principles to create AI-generated self-critiques to enhance the models autonomously. During the self-critique process, however, only a single rule is randomly chosen to scrutinize the existing response. Sun et al. (2023) improves on this approach by incorporating 16 manually-devised guiding principles that entail broader domains and more specific criteria, such as candorness, step-by-step justifications, and multi-faceted answers. By broadening the range of topics, they allow the language model to decide which principles to adhere to given user queries. However, these approaches are resource-intensive and demand significant human labor, as they necessitate explicitly predefined guiding principles.

Prior work has recognized the importance of guiding LLM

generations using principles situated in the particular context at hand, such as allowing users to formulate principles that steer the conversation (Petridis et al., 2024b). However, relying solely on human interactions to provide such context-situated guidance is challenging to scale. In Chen et al. (2024), strong LLMs are used to discover principles for a weak LLM. In this red-teaming approach, both a stronger LLM and an initial *bad* response are necessary, thus difficult to generalize. Petridis et al. (2024a) also introduces a method for learning a collection of constitutional principles given a cluster of training data. The training is conducted on various clusters of data, resulting in different sets of principles. At inference time, input queries are then directed to different principles based on their similarity to the centroids of the training clusters. Similarly, OpenAI o1 models (Jaech et al., 2024) utilize a technique entitled Deliberative Alignment (Guan et al., 2025), which teaches LLMs to explicitly reason through safety specifications before producing an answer, but their approach mainly seeks to align and train a downstream model.

In contrast, our method customizes the principles for each individual input query, rather than basing them on a set of undesirable responses or a cluster of training data. This ensures that the principles are not generalized but specifically tailored to each unique input query, making our constitutional principles more precise. Our framework is also more versatile and not restricted to supervised fine-tuning. As demonstrated in §4, SPRI can effortlessly extend to complex tasks that require significant human oversight.

**Learning from Feedback.** To align AI systems with human preferences and values, researchers have explored using human feedback to direct the behaviors of language models (Kirk et al., 2023a). This includes efforts to incorporate human feedback in the pretraining (Korbak et al., 2023) and supervised fine-tuning phases (Hancock et al., 2019; Liu et al., 2024), integrate human feedback through reinforcement learning either directly (Stiennon et al., 2020; Bai et al., 2022a; Bakker et al., 2022; Ouyang et al., 2022; Liu et al., 2022) or indirectly (Zhou et al., 2021; Korbak et al., 2023), as well as prompt engineering (Jin et al., 2022; Zhao et al., 2021; Askell et al., 2021).

However, human feedback is expensive and laborious to collect (Lee et al., 2024). Other works have therefore resorted to using machine-generated feedback for improving the model outputs (Bai et al., 2022b; Yang et al., 2022; Lee et al., 2024; Fu et al., 2024; Cui et al., 2024; Ramji et al., 2024). Self-refine (Madaan et al., 2023) explored the approach where an LLM first generates an initial response, then reviews and evaluates its own output and iteratively improves it based on the self-generated feedback until it meets a certain quality standard. Ramji et al. (2024) adapted this approach for smaller language models, which have more

limited self-critique and refinement abilities, by enabling them to improve their responses based on feedback from external quality metrics. Bai et al. (2022b) proposed RLAIF, a method that combines LLM- and human-labeled preferences to optimize for both helpfulness and harmlessness in a response simultaneously. Dubois et al. (2023) further introduced the AlpacaFarm simulator, which allows for a large-scale collection of pairwise feedback for model outputs using API-based LLMs. Our approach differs from these methods by focusing on refining the principles tailored to each input, in addition to refining the outputs. These principles are then used to guide the generation of responses for each *corresponding* input and serve as the criteria for critiquing and improving the responses.

## 3. SPRI: A Scalable Alignment Framework with Minimal Human Oversight

We present SITUATED-PRINCIPLES (SPRI), a framework that generates context-situated principles to align LLMs while minimizing human oversight. The framework relies on two ingredients: a base model $\mathcal{M}$ and a critic model $\mathcal{C}$. An overview of SPRI is shown in Figure 2. To generate an aligned response, SPRI goes through two steps: during the ***first*** stage, $\mathcal{M}$ takes in the user's input $T$ and generates a set of principles customized to $T$ through a series of critique-refinement loops with $\mathcal{C}$; then in the ***second*** stage, the generated principles are fed into $\mathcal{M}$ to guide its response. These principles also serve as criteria to provide feedback on the generated responses for improvement. We provide the pseudo-code algorithms in Appendix §A.

**Stage I: Synthesizing Context-Situated Principles.** Based on a user's input $T$, the objective of the first step is to generate guiding principles tailored to $T$. Given $T$, the base model $\mathcal{M}$ is prompted with $P_{\text{principle-gen}}$ to produce an initial set of principles, $K_0$, as follows:

$$K_0 = \mathcal{M}(T \oplus P_{\text{principle-gen}} \oplus S), \qquad (1)$$

where $\oplus$ denotes concatenation and $P_{\text{principle-gen}}$ is a prompt instructing the model to generate principles (see Appendix §C). A set of seed *(instruction, principle)* tuples, denoted as $S$, can also be provided as few-shot examples for the model to better grasp the essence of desired principles. We note that the provision of seed examples is optional: this initial principle-generation phase can be rendered under a zero-shot setting.

As the next step, we need to determine the adequacy of $K_0$ and assess whether it is suitable for guiding the response to $T$. We use the critic model $\mathcal{C}$ to yield feedback on $K_0$:

$$\text{Feedback}_{K_0} = \mathcal{C}(Eval_{\text{principle}} \oplus T \oplus K_0). \qquad (2)$$

Here, $Eval_{\text{principle}}$ is a chain-of-thought (Wei et al., 2022) style evaluation prompt in the format of direct assessment (Kim et al., 2024) that instructs $\mathcal{C}$ to produce both qualitative feedback and a numerical score (on a 1 to 5 Likert scale). The feedback is fed back into the base model $\mathcal{M}$, prompting it to refine the principles:

$$K_i = \mathcal{M}(P_{\text{principle-refine}} \oplus T \oplus K_{i-1} \oplus \text{Feedback}_{K_{i-1}}), \quad (3)$$

where $P_{\text{principle-refine}}$ is a prompt instructing the model to refine principles based on feedback. This iterative critique-refinement process continues until the principles receive a desired score of at least 4 or a maximum of four iterations is reached. We denote the final set of principles deemed suitable to guide the response to $T$ as $K_{\text{final}}$.

**Stage II: Generating Responses Guided by Synthesized Principles.** We use the established principles $K_{\text{final}}$ to guide $\mathcal{M}$'s response to $T$. The initial response generation process can be expressed as:

$$R_0 = \mathcal{M}(T \oplus P_{\text{response-gen}} \oplus K_{\text{final}}), \quad (4)$$

where $P_{\text{response-gen}}$ is a prompt that instructs $\mathcal{M}$ to respond. $R_0$ is then examined by the critic model $\mathcal{C}$ for feedback, with the principles $K_{\text{final}}$ being the rubrics:

$$\text{Feedback}_{R_0} = \mathcal{C}(Eval_{\text{response}} \oplus T \oplus K_{\text{final}} \oplus R_0). \quad (5)$$

Similar to Stage I, $Eval_{\text{response}}$ is a direct assessment prompt that elicits feedback and a score from $\mathcal{C}$. If the evaluation score is below 4 or the maximum number of iterations is not reached, the feedback is passed back to the base model $\mathcal{M}$ to iteratively refine its response:

$$R_i = \mathcal{M}(P_{\text{response-refine}} \oplus T \oplus R_{i-1} \oplus \text{Feedback}_{R_{i-1}}). \quad (6)$$

Here, $P_{\text{response-refine}}$ is a prompt asking the model to refine the response based on feedback. We denote the final refined response as $R_{\text{final}}$. By iteratively refining both the guiding principles and the response, SPRI ensures that $R_{\text{final}}$ aligns closely with the user's input $T$ and the generated principles $K_{\text{final}}$ with minimal to no human intervention. While the critique-refine process in Stage II of SPRI shares similarities with self-refine (Madaan et al., 2023), it is distinctly guided by context-situated principles $K_{\text{final}}$ generated from Stage I. SPRI is easy to scale and can be dynamically adapted to diverse user inputs and tasks: not only can it extrapolate to complex tasks such as providing emotional support (§4.1) or performing instance-specific evaluation (§4.2), but it also performs well on providing training data for large-scale alignment (§5).

# 4. SPRI for Complex Principles

We examine the effectiveness of SPRI on complex real-world tasks, one where LLMs are shown only to be successful if provided with complex, expert-curated principles in the prompt (Zhan et al., 2024), another on a larger benchmark where manually curated situation-specific rubrics are necessary (Kim et al., 2025). We show that SPRI generates effective principles for complex tasks in the former (§4.1), and also generates evaluation rubrics for instance-level assessment in the latter (§4.2). We detail the token usage and computational costs of SPRI compared to other methods in Appendix §B, and provide example SPRI-generated principles in Appendix §J.

## 4.1. Can SPRI Guide Cognitive Reappraisals?

We explore how SPRI can be applied to facilitate *cognitive reappraisals*, a strategy widely recognized by psychology practitioners that aims to promote long-term mental well-being for an individual (Gross, 1998; Gross & John, 2003; Waugh et al., 2016). Recently, Zhan et al. (2024) showed that complex principles crafted by professional psychologists used in LLM prompts enables the models to perform this complex task. An oracle principle is used for each individual appraisal dimension (refer to Appendix §E for details). This is an ideal testbed for SPRI to dynamically generate complex context-specific principles to guide the elicitation of reappraisal responses. By developing a unique set of principles *from scratch* for each individual user query, we show performance comparable to those guided by oracle principles while minimizing human supervision.

**Data.** We evaluate on the same dataset from Zhan et al. (2024). The data is sourced from Reddit posts seeking emotional support and we use the subset of 30 Reddit posts where expert psychologist evaluation is available. The average post length is 170.5 tokens (SD = 99.2).

**Baselines.** We first explore two **principle-free methods**, including **1)** *vanilla*, a weak baseline in which a generic prompt "*help the narrator of the text reappraise the situation*" is used to elicit a straightforward reappraisal response from the language model. **2)** *self-refine* (Madaan et al., 2023), which builds on the vanilla prompt by incorporating a single feedback repeatedly six times: "*please revise the reappraisal response to help the narrator reappraise the situation better.*" This serves as a baseline for refinement without guidance. Additionally, we also experiment with an **oracle-informed method** that leverages predefined reappraisal principles in the prompts: **3)** *+oracle*, where we provide the language model with the detailed, expert-crafted reappraisal constitutional principles from RESORT. This offers insight into how SPRI performs relative to systems with access to expert-designed guidelines.

**SPRI Method.** To increase the stability of the principle generation process, we provide SPRI with a single oracle RESORT constitution as the seed example.

**Evaluation & Criteria.** We adopt the evaluation schema from Zhan et al. (2024), which is comprised of 4 criteria that extensively assess the quality of reappraisals generated by LLMs, namely: *1) Alignment with Reappraisal Constitutions*, which assesses whether the reappraisal response adheres to the oracle constitutions specified by Zhan et al. (2024). Responses are rated from 1 to 10, with 1 being *"Least Aligned"* and 10 being *"Most Aligned"*. *2) Empathy*, which evaluates whether the reappraisal response shows empathy towards the narrator of the Reddit post on a scale from 1 to 5, with 1 being *"Least Empathetic"* and 5 indicating *"Most Empathetic"*. We consider these two metrics the key to evaluating reappraisals. In addition, we also look at the *3) Harmfulness* of the response, checking whether the response contains any unethical or harmful content, with options being *"Harmful"* (1) and *"Not Harmful"* (0). Finally, *4) Factuality* measures whether the response is factually consistent in relation to the given Reddit Post, with options *"Yes"* (1), *"Minor Error"* (0.5), and *"No"* (0). We leave the results for these two dimensions in Appendix §G.

We carry out automatic evaluation on all reappraisal responses elicited using GPT-4-0613, using the method from (Zhan et al., 2024) which showed strong correlation with evaluation results conducted by professional psychologists.

**Experimental Setup.** We experiment with a comprehensive suite of state-of-the-art LLMs, including GPT-4o-mini (Hurst et al., 2024), Llama-3.1-70B-Instruct and Llama-3-8B-Instruct (Dubey et al., 2024), as well as Mixtral-8x7B-Instruct-v0.1 (Jiang et al., 2024). In the SPRI method, these models act as the base model $\mathcal{M}$. We employ Prometheus-2-8x7B (Kim et al., 2024), a mixture-of-experts model developed specifically for the task of giving feedback, as the critic model $\mathcal{C}$ for all SPRI experiments. We set the temperature $T = 0.7$ for model inferencing.

**Results.** We show the results in Table 1.[2] First, we note that oracle-informed approaches significantly outperform principle-free baselines. Notably, incorporating oracle principles in the prompt (oracle principles) increases models' performance over vanilla and self-refine methods by an average of 11.3% and 16.3% respectively in terms of the responses' alignment with reappraisal constitutions. On the other hand, **SPRI consistently outperforms methods that lack access to oracle principles both in terms of reappraisal alignment and perceived empathy, even though it only utilizes a single seed principle.** Specifically, we obtain an average improvement of 6.1% in alignment and 8.4% in empathy over our strongest vanilla baseline. Moreover, our SPRI approach also significantly surpass the

---

[2]Zhan et al. (2024) presented two strategies to incorporate the oracle principles, and we report the better one here. Please see Appendix §G Figure 8 for the full results with both strategies.

self-refine method by as much as 11.0% in alignment and 12.1% in empathy. These results suggest that tailoring context-situated principles can achieve performance comparable to those with oracle guidance, even for a task as complex as offering psychologically grounded emotional support.

### 4.2. Can SPRI Generate Fine-Grained Rubrics?

We further investigate SPRI's capability to handle case-by-case nuances by examining its ability to generate fine-grained evaluation rubrics for each individual instance. We utilize BiGGen Bench (Kim et al., 2025), an extensive benchmark designed to assess the performance of LLMs across a variety of tasks using language models. BiGGen Bench stands out due to its use of instance-specific evaluation rubrics, each meticulously curated to ensure detailed and contextually rich assessments. We detail the BiGGen Bench dataset in Appendix §F. While these human-crafted criteria allow for a fine-grained analysis of models' performance on *each individual case*, the manual creation of such detailed rubrics is both labor-intensive and time-consuming. To mitigate this bottleneck, we propose leveraging SPRI to automate the rubric generation process. Specifically, **we hypothesize that LLMs, when guided by the SPRI framework, can produce evaluation rubrics from scratch that align closely with human-annotated ones in quality and contextual specificity for each individual evaluation instance**.

**Data.** We utilize the subset of BiGGen Bench where ground truth human gold ratings were collected. Specifically, we focus on 8 different capabilities, namely *instruction-following*, *refinement*, *theory of mind*, *grounding*, *reasoning*, *planning*, *tool usage*, and *safety*. This results in a total of 2,780 *(response, gold rating)* pairs, spanning across 695 evaluation instances.

**Baselines.** Similar to the setup in §4.1, we first experiment with eliciting evaluation rubrics using **instance-agnostic methods**, namely **1)** *vanilla*, a weak baseline where we use a generic prompt *"How well does the response address the instruction? Please rate on a scale of 1 to 5, where 1 stands for 'not at all' and 5 stands for 'perfectly'"* to evoke a pristine judgment from the language model. **2)** *self-refine* (Madaan et al., 2023), where the vanilla prompt is formulated as repeated feedback, a baseline for refinement *without* guidance. Please note that we do not set a "sufficient" stopping criteria here, but instead only impose a max iteration of 6, as in practice we find that the model tends to rate all of its responses sufficient with no need for refinement. **3)** *MT-Bench rubric* (Zheng et al., 2023), a coarse-grained criteria that assesses the quality of the response from aspects including helpfulness, relevance, accuracy, depth, creativity,

*Table 1.* Evaluation results (in average scores) for reappraisal responses. We report statistical significance (with $p < 0.05$) using pair-wise t-tests against both the vanilla (marked with *) and self-refine (marked with †) baselines. Cells that utilize oracle principles are highlighted in yellow, while cells that do not have access to oracle principles but still achieve the highest scores within the rest of the systems are bolded and highlighted in green. For the full results, see Appendix §G Figure 8.

| | GPT-4o-mini | | Llama-3.1-70B-Instruct | | Llama-3-8B-Instruct | | Mixtral-8×7B-Instruct | |
|---|---|---|---|---|---|---|---|---|
| | **Alignment ↑** | **Empathy ↑** | **Alignment ↑** | **Empathy ↑** | **Alignment ↑** | **Empathy ↑** | **Alignment ↑** | **Empathy ↑** |
| | Scale of 10 | Scale of 5 | Scale of 10 | Scale of 5 | Scale of 10 | Scale of 5 | Scale of 10 | Scale of 5 |
| vanilla | 7.90 | 4.50 | 7.77 | 4.43 | 7.10 | 3.90 | 7.53 | 4.50 |
| self-refine | 7.73 | 4.53 | 7.50 | 4.27 | 7.20 | 4.07 | 6.60 | 3.90 |
| SPRI | **8.00**† | **4.73** | **8.17***† | **4.77***† | **7.90***† | **4.47***† | **8.03***† | **4.77***† |
| oracle principles | 8.67*† | 4.80*† | 8.53*† | 4.20 | 8.33*† | 4.30* | 8.17 | 4.07 |

*Table 2.* Results for BiGGen Bench, measured using Pearson's correlation to ground truth human labels. Evaluation carried out *without* the use of reference answers. Cells that utilize oracle rubrics are highlighted in yellow, whereas cells that do not have access to oracle rubrics but still achieve the highest scores within the rest of the systems are bolded and highlighted in green. See Appendix §H Table 9 for the full results.

| | GPT-4o mini | Llama-3.1-70B Instruct | Mixtral-8x7B Instruct | Prometheus-2 8x7B |
|---|---|---|---|---|
| vanilla | 0.377 | 0.386 | **0.307** | 0.311 |
| self-refine | 0.397 | 0.260 | 0.110 | 0.297 |
| MT-Bench rubric | 0.416 | 0.421 | 0.273 | 0.289 |
| FLASK rubric | 0.358 | 0.360 | 0.277 | 0.294 |
| SPRI | **0.472** | **0.480** | 0.288 | **0.333** |
| oracle rubrics | 0.550 | 0.556 | 0.367 | 0.386 |

and the level of detail. **4)** *FLASK rubric* (Ye et al., 2024), a set of domain-specific criteria that covers areas like logical robustness, factuality, commonsense understanding, comprehension, insightfulness, meta-cognition, and harmlessness. We further experiment with an **oracle-informed method**: **5)** *oracle rubrics*, where the human-crafted ground truth criteria from Kim et al. (2024) are provided to evaluator LMs as rubrics.

**SPRI Methods.** To increase the stability of the principle generation process, we augment SPRI with 3 instance-rubric pairs from BiGGen Bench as seed examples for each capability. Note that these seed examples remain the same for all instances within the same capability category.

**Experimental Setup.** We experiment with a comprehensive suite of state-of-the-art LLMs, including GPT-4o-mini, Llama-3.1-70B-Instruct, Mixtral-8x7B-Instruct-v0.1, as well as Prometheus-2-8x7B. In the SPRI methods, these models act as the base model $\mathcal{M}$. We employ Prometheus-2-8x7B as the critic model $\mathcal{C}$ for all SPRI experiments.

**Evaluation.** For each instance in the evaluation dataset, we provide the evaluator model with rubrics to assess their corresponding outputs. We use the template from Prometheus (Kim et al., 2024) to prompt the evaluator model. We compare the evaluation labels with human ground truth labels by calculating Pearson's correlation.

Note that in the BiGGen Bench dataset, each instance is also accompanied by a reference answer. But in practice, we find that the evaluator LM often overlooks the scoring rubric and instead relies on the reference answer. To ablate the influence of the scoring rubrics in our experiments, we *don't* use reference answers throughout the evaluation.

**Results.** We provide the average Pearson's correlation to ground truth human labels in Table 2. Similar to the results from cognitive reappraisals (§4.1), systems with access to oracle rubrics outperform methods employing instance-agnostic rubrics by a considerable margin. The coarse-grained MT-Bench rubric leads to a moderate performance among the instance-agnostic baselines, whereas the domain-specific FLASK rubric often lags behind. **Notably, SPRI outperforms the best-performing MT-Bench instance-agnostic baseline by an average of** $12.1\%$**, while only relying on** $3$ **oracle rubrics as seeds.** Although oracle rubrics exceeds SPRI in performance, the difference is relatively small, leading to an average margin of only 0.07 in Pearson's correlation across all models. These results, combined with the findings in §4.1, underscore the potential of SPRI in enhancing the LLMs' robustness for tasks that require complex principles and guidance.

### 4.3. Ablation Study

To better tease apart and analyze the success of SPRI, we study the impact of seed examples provided in the initial principle generation stage. We first remove seed examples from the SPRI pipeline. We denote this approach by -seed=[none]. In order to further demonstrate the robustness of SPRI, we insert generic principles (shown in Appendix §D Figure 3) as seed examples, and denote this

*Table 3.* Ablation for SPRI on reappraisal responses (measured by their responses' alignment to reappraisal constitutions), and BiGGen Bench rubric generation. Reappraisal responses where the ratings are significantly *worse* than either of the `vanilla` and `self-refine` baselines are shaded.

| | REAPPRAISAL ALIGNMENT | | | | RUBRIC GENERATION | | | |
|---|---|---|---|---|---|---|---|---|
| | GPT-4o mini | Llama-3.1-70B Instruct | Llama-3-8B Instruct | Mixtral-8x7B Instruct | GPT-4o mini | Llama-3.1-70B Instruct | Mixtral-8x7B Instruct | Prometheus-2 8x7B |
| SPRI | **8.00**[†] | **8.17**[*†] | **7.90**[*†] | **8.03**[*†] | **0.472** | **0.480** | **0.288** | 0.333 |
| -seed=[none] | 7.67[*] | 7.77 | 7.73[*†] | 7.60[†] | 0.410 | 0.410 | 0.245 | 0.297 |
| -seed=[default_principles] | 7.67 | 7.87[†] | 7.70[*†] | 7.57[†] | 0.404 | 0.391 | 0.238 | **0.336** |
| default_principles only | 2.13[*†] | 6.47[*†] | 6.07[*†] | 2.80[*†] | 0.176 | 0.055 | 0.260 | 0.308 |

modification as `-seed=[default_principles]`. We show-case the results in Table 3. Removing seed examples entirely leads to an average performance degradation of 4.13% in alignment for reappraisals and 13.37% in Pearson's correlation for rubric generation. On the other hand, substituting the default principles as seeds leads to a similar average performance decrease of 4.01% in alignment and 12.35% in Pearson's correlation for rubric generation. These results highlight the robustness of SPRI to seed examples in the initial principle-generation stage, as our default principles are neither relevant to the tasks we evaluate nor fit to the instances we aim to provide guidance with.

Additionally, to better understand the influence of the seed principles on SPRI, we also experiment with a separate condition ***default_principles only***, where we randomly select one of the six default principles and include it as both the final guiding principle for eliciting reappraisals and the final rubrics for evaluating instances. This helps ablate the influence of the default principles within the SPRI pipeline, as they are unrelated to both the reappraisal task and the context at hand. As shown in Table 3, utilizing default principles alone in the prompt to guide LLMs for the task of cognitive reappraisals leads to an average performance decrease of 45.62% compared to SPRI, and this degradation is most observed for `GPT-4o-mini` and `Mixtral-8x7B-Instruct`. In terms of instance-specific evaluation, employing default principles alone led to the most performance degradation for the more capable models `GPT-4o-mini` and `Llama-3.1-70B-Instruct` on this task, where their Pearson's correlation score go down by 62.7% and 88.5% respectively compared to SPRI. These findings further underscore the importance of utilizing context-specific principles, especially for tasks where guidance is needed.

## 5. Can SPRI Generate Large-Scale Alignment Data for Supervised Fine-Tuning?

Finally, we apply SPRI to a more general setting: generating large-scale synthetic data for supervised fine-tuning (SFT). Through evaluating language models fine-tuned on our synthetically generated data, we indirectly assess the capability of SPRI. Intrinsically, SPRI's context-situated principles allow for a deeper ability to reject misleading claims — as exhibited in Appendix §J.3, when provided with questions that don't have a definite answer (e.g., *"Is it true that if you don't exercise your body will become weaker?"*), SPRI often generates guiding principles that asks the response to focus on both sides of the question. Based on the nature of SPRI, we hypothesize that SPRI would perform best on benchmarks that measure the rejection of falsehoods, whilst maintaining the performance in the knowledge as well as problem-solving domains.

### 5.1. Task Formulation

Let $\phi(x)$ be the pipeline we generate responses with, and let $\mathcal{F}_\theta$ be a model that we want to align. We are interested in aligning $\mathcal{F}_\theta$ using the data $\phi(x)$ produces. To this end, given an instruction-following dataset $D$ that is composed of prompt-response pairs $D = \{(p_1, r_1), (p_2, r_2), ..., (p_n, r_n)\}$, we aim to produce corresponding aligned responses conditioned on the prompts: $\{\phi(p_1), \phi(p_2), ..., \phi(p_n)\}$. Subsequently, we construct a new dataset $D_\phi$, which consists of the original prompts paired with their corresponding aligned responses. We then train $\mathcal{F}_\theta$ on $D_\phi$ by optimizing its weights $\theta$, resulting in a trained model $\mathcal{F}_{\theta^*}$. We measure the performance of $\mathcal{F}_{\theta^*}$ as an indicator of the quality of $D_\phi$.

### 5.2. Experimental Setup

**Data.** To examine the generalizability of SPRI, we carry out experiments on two different instruction-tuning datasets $D$, namely `Dolly` (Conover et al., 2023) and `MixInstruct` (Jiang et al., 2023). `Dolly` contains around 15k manually curated prompt-response pairs, whereas `MixInstruct` consists of 110k examples where the responses are primarily sourced from `GPT-3.5-turbo` and `GPT-4`. We randomly split `Dolly` into a 10k/2k split for training and validation. For `MixInstruct`, we randomly select 50k examples from its training set and 2k examples from its validations set.

*Table 4.* Performance of supervised fine-tuned models on TruthfulQA (Lin et al., 2022).

| | Llama-3.1-8B | | Llama-3.1-8B-Instruct | | Mistral-7B-v0.3 | | Mistral-7B-v0.3-Instruct | | Gemma-2-9B | | Gemma-2-9B-it | |
|---|---|---|---|---|---|---|---|---|---|---|---|---|
| | Dolly | MixInstruct | Dolly | MixInstruct | Dolly | MixInstruct | Dolly | MixInstruct | Dolly | MixInstruct | Dolly | MixInstruct |
| oracle response | 41.62% | 51.94% | 46.75% | 49.28% | 40.42% | 50.90% | 42.87% | 49.64% | 44.81% | 51.21% | 47.11% | 57.48% |
| direct response | 51.48% | 50.82% | 50.94% | 50.99% | 47.16% | 52.64% | 50.89% | 55.09% | 53.82% | 53.94% | 57.97% | 57.73% |
| self-instruct | 51.07% | 52.02% | 49.46% | 50.76% | 46.62% | 51.87% | 50.44% | 52.81% | 52.43% | 52.85% | 56.26% | 54.70% |
| self-align | 54.56% | 54.97% | 52.52% | 51.96% | 48.86% | 53.95% | 54.44% | 56.85% | 54.02% | 51.70% | 58.34% | 55.11% |
| self-refine | 53.76% | 55.11% | 52.11% | 50.20% | 49.40% | 53.15% | 52.35% | 54.69% | 55.01% | 53.93% | 58.86% | 58.36% |
| seed principles | 53.63% | 53.83% | 50.46% | 52.90% | 50.89% | 54.24% | 52.42% | 56.53% | 53.48% | 52.22% | 57.96% | 58.24% |
| SPRI | **55.92%** | **56.08%** | **54.69%** | **55.41%** | **51.85%** | **55.63%** | 56.43% | 57.99% | **55.72%** | **56.48%** | **62.62%** | **59.75%** |
| off-the-shelf | 45.03% | | 53.02% | | 42.54% | | **66.11%** | | 45.39% | | 60.47% | |
| post-trained | 53.02% | | — | | 66.11% | | — | | 60.47% | | — | |

**Baseline Methods.** We experiment with a variety of baselines, including **1)** *oracle response*, where we fine-tune directly on the oracle responses provided in the datasets. **2)** *direct response*, in which we collect responses by asking the base model $\mathcal{M}$ to directly respond to the instructions for each instance in the dataset. **3)** *self-instruct*, where we elicit responses from $\mathcal{M}$ by relying on a few-shot prompt with 11 *(input, output)* example pairs from Wang et al. (2023). **4)** *topic-guided red-teaming*, a prompt from Sun et al. (2023), in which a set of 16 general rules as well as few-shot examples demonstrating how to utilize these rules in a chain-of-thought (Wei et al., 2022) fashion are used to elicit responses. **5)** *self-refine* (Madaan et al., 2023), where we ask the base model $\mathcal{M}$ to critic and refine its own response. During critiquing, we ask the model to provide feedback followed by an integer assessment score from 1 to 5. We iterate the critique-refine process until a minimal assessment score of 4 is met or the maximum number of iterations of 4 is reached. In addition, we also experiment with **6)** *seed principles*, where we utilize the 6 default principles (shown in Appendix §D Figure 3) as the guiding principles for the model to generate responses. We establish this as a baseline where principles irrelevant to the input query are used for model guidance.

**SPRI Method.** We supply SPRI with the 6 *Question–Principle* pairs shown in Figure 3 as seed examples during the initial principle generation phase.

**Models and Setup.** We use `Llama-3-70B-Instruct` (Dubey et al., 2024) as our base model $\mathcal{M}$ across all methods, and we employ `Prometheus-2-8x7B` as the critic model $\mathcal{C}$ in SPRI. We set the temperature value for all model generations to 0.7, top $k$ to 50, top $p$ to 0.95. We also restrict the maximum tokens of generation to 256.

We finetune with LoRA (Hu et al., 2022), and we compute the loss on responses only. For base (i.e., non-instruction-tuned) models, we use the Alpaca format template (Taori et al., 2023) for training; for instruction-tuned models, we fine-tune them on their own chat templates. We save the best model checkpoint at validation loss as the final model. All

our fine-tuning experiments are carried out on 3 NVIDIA A100 40GB GPUs.

### 5.3. Results

We evaluate the performance of fine-tuned models on several benchmarks, namely TruthfulQA (Lin et al., 2022), MUSR (Sprague et al., 2024), GPQA (Rein et al., 2024), BBH (Suzgun et al., 2023), MMLU-Pro (Wang et al., 2024), and Hellaswag (Zellers et al., 2019). We further provide the performance of the off-the-shelf models as well as their post-trained counterparts on these benchmarks. As shown in Table 4, **SPRI consistently outperforms the off-the-shelf model as well as other synthetic response generation methods on the TruthfulQA dataset.** In particular, fine-tuning base models using SPRI leads to the most notable gains on the benchmark, surpassing the off-the-shelf models' performance by an average of 24.76% and models fine-tuned using oracle responses by an average of 19.09%. While already instruction-tuned models benefit from smaller gains with SPRI, their performance still exceeds all baseline methods. In particular, `Llama-3.1-8B-Instruct` outperforms its off-the-shelf and oracle-response fine-tuned counterparts' performance on TruthfulQA by a margin of 3.83% and 14.71% respectively.

We further provide the results from SFT on other benchmarks in Appendix §I Tables 10 and 11. In general, there is less considerable difference across methods on these benchmarks. While we observe the effect of alignment tax (Askell et al., 2021; Ouyang et al., 2022) where post-trained models are weaker than base counterparts on benchmarks such as MUSR and Hellaswag, this effect is less observed for models fine-tuned using SPRI. Instead, SPRI's performance is often comparable to the best-performing method on MUSR, GPQA, BBH, MMLU-Pro, and Hellaswag. These results highlight the effectiveness of SPRI on aligning models, particularly in terms of truthfulness.

# 6. Conclusion

We introduce SPRI, a framework that produces context-situated principles tailored to each input query at hand. Through a series of extensive evaluations on tasks including cognitive reappraisals, instance-specific rubric generation, and generating synthetic data for SFT, we demonstrate the effectiveness of SPRI in guiding responses. By dynamically generating principles in real time with minimal or no human effort, SPRI addresses key limitations of prior approaches that relied on generic, static principles. Our results show that SPRI not only matches expert-level performance in highly specialized tasks but also enhances alignment with human judgment and improves synthetic data generation for model fine-tuning. This work underscores the potential of SPRI to enable more adaptable, context-aware, and scalable alignment strategies for LLMs, paving the way for broader applicability in tasks requiring nuanced human oversight and guidance.

## Acknowledgements

We sincerely thank the reviewers for their constructive feedback on this work. We also thank Heloisa Candello for her valuable input to the default principles used in this paper. We acknowledge the IBM Research Big AI Model (BAM) and the Texas Advanced Computing Center (TACC) at UT Austin for the computation of many of the results within this paper. This work was partially supported by NSF grants IIS-2107524, IIS-2145479, and Good Systems, a UT Austin Grand Challenge to develop responsible AI technologies.

## Impact Statement

This paper presents work whose goal is to advance the field of Machine Learning. While there are many potential societal consequences of our work, we feel that this statement suffices as a broad impact assessment.

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

# A. Pseudo-code for SPRI

---

**Algorithm 1** Pseudo-code for SPRI

---

**Require:** user input $T$, base language model $\mathcal{M}$, critic language model $\mathcal{C}$, seed examples $S$ (optional), prompts $\{P_{\text{principle-gen}}, P_{\text{principle-refine}}, P_{\text{response-gen}}, P_{\text{response-refine}}\}$, evaluation prompts $\{Eval_{\text{principle}}, Eval_{\text{response}}\}$, max iterations $n_{\max}$, desired score threshold $\tau$.

   STAGE I: SYNTHESIZING CONTEXT-SITUATED PRINCIPLES

1: Initialize $\mathcal{M}, \mathcal{C}$
2: $K_0 = \mathcal{M}(T \oplus P_{\text{principle-gen}} \oplus S)$                                        {Generate the initial principles $K_0$}
3: Reset $\mathcal{M}$
4: **for** $i = 1$ to $n_{\max}$ **do**
5:    $\text{Feedback}_{K_{i-1}} = \mathcal{C}(Eval_{\text{principle}} \oplus T \oplus K_{i-1})$                   {Evaluate $K_{i-1}$ using the critic model $\mathcal{C}$}
6:    Extract score from $\text{Feedback}_{K_{i-1}}$
7:    **if** score $\geq \tau$ **then**
8:       $K_{\text{final}} = K_{i-1}$; **break**
9:    **end if**
10:   $K_i = \mathcal{M}(P_{\text{principle-refine}} \oplus T \oplus K_{i-1} \oplus \text{Feedback}_{K_{i-1}})$            {Refine principles $K_{i-1}$}
11:   Reset $\mathcal{M}, \mathcal{C}$
12: **end for**
13: **if** score $< \tau$ after $n_{\max}$ iterations **then**
14:   $K_{\text{final}} = K_{n_{\max}}$
15: **end if**
   STAGE II: GENERATING RESPONSES GUIDED BY SYNTHESIZED PRINCIPLES
16: $R_0 = \mathcal{M}(T \oplus P_{\text{response-gen}} \oplus K_{\text{final}})$                          {Generate the initial response $R_0$}
17: Reset $\mathcal{M}$
18: **for** $i = 1$ to $n_{\max}$ **do**
19:   $\text{Feedback}_{R_{i-1}} = \mathcal{C}(Eval_{\text{response}} \oplus T \oplus K_{\text{final}} \oplus R_{i-1})$        {Evaluate $R_{i-1}$ using the critic model $\mathcal{C}$}
20:   Extract score from $\text{Feedback}_{R_{i-1}}$
21:   **if** score $\geq \tau$ **then**
22:      $R_{\text{final}} = R_{i-1}$; **break**
23:   **end if**
24:   $R_i = \mathcal{M}(P_{\text{response-refine}} \oplus T \oplus R_{i-1} \oplus \text{Feedback}_{R_{i-1}})$           {Refine response $R_{i-1}$}
25: **end for**
26: **if** score $< \tau$ after $n_{\max}$ iterations **then**
27:   $R_{\text{final}} = R_{n_{\max}}$
28: **end if**
29: **return** Final guiding principles $K_{\text{final}}$ and response $R_{\text{final}}$

---

# B. Number of Tokens & Computational Costs Induced by SPRI

We provide here the token usage and computational costs of SPRI versus other methods for each task in our experiments. We report the average model calls & input/output token usage per response for the base and critic models, as well as the estimated total cost to carry out an entire task. We estimate the cost using OpenAI's API pricing for GPT and TogetherAI's pricing for open-source models.

## B.1. Cognitive Reappraisal

*Table 5.* Computational cost and token usage using different methods to carry out the task of cognitive reappraisals (§4.1). Here, we use `GPT-4o-mini` as the Base Model, and `Prometheus-2-8x7B` as the Critic Model.

| | Average | BASE MODEL | | | CRITIC MODEL | | |
|---|---|---|---|---|---|---|---|
| | Model Calls | Input Tokens | Output Tokens | Total Cost | Input Tokens | Output Tokens | Total Cost |
| `vanilla` | 1 | 299 | 94 | $0.003 | – | – | – |
| `self-refine` | 6 | 2,106 | 465 | $0.018 | – | – | – |
| `oracle` | 6 | 4,280 | 1,421 | $0.045 | – | – | – |
| `SPRI` | 4.5 | 639 | 220 | $0.007 | 1,537 | 281 | $0.033 |

## B.2. Instance-Specific Rubric Evaluation

*Table 6.* Computational cost and token usage to generate instance-specific evaluation rubrics for LLM-as-a-judge (§4.2). We use `GPT-4o-mini` as the Base Model, and `Prometheus-2-8x7B` as the Critic Model.

| | Average | BASE MODEL | | | CRITIC MODEL | | |
|---|---|---|---|---|---|---|---|
| | Model Calls | Input Tokens | Output Tokens | Total Cost | Input Tokens | Output Tokens | Total Cost |
| `vanilla` | 1 | 568 | 99 | $0.403 | – | – | – |
| `self-refine` | 6 | 4,147 | 619 | $2.762 | – | – | – |
| `MT-Bench rubric` | 1 | 469 | 200 | $0.530 | – | – | – |
| `FLASK rubric` | 1 | 636 | 103 | $0.437 | – | – | – |
| `oracle` | 1 | 707 | 105 | $0.469 | – | – | – |
| `SPRI` | 4.9 | 1,720 | 317 | $1.247 | 2,642 | 282 | $4.877 |

## B.3. SFT

*Table 7.* Computational cost and token usage for generating large-scale alignment data for SFT (§5). We use `Llama-3-70B-Instruct` as the Base Model, and `Prometheus-2-8x7B` as the Critic Model. The estimate is based on using `Dolly` as the starting instruction-tuning dataset.

| | Average | BASE MODEL | | | CRITIC MODEL | | |
|---|---|---|---|---|---|---|---|
| | Model Calls | Input Tokens | Output Tokens | Total Cost | Input Tokens | Output Tokens | Total Cost |
| `direct response` | 1 | 113 | 61 | $1.9 | – | – | – |
| `self-instruct` | 1 | 1,046 | 99 | $12.2 | – | – | – |
| `self-align` | 1 | 1,116 | 143 | $13.4 | – | – | – |
| `self-refine` | 2.1 | 139 | 69 | $2.2 | – | – | – |
| `SPRI` | 5.0 | 1,077 | 167 | $13.3 | 1,363 | 258 | $11.5 |

### B.4. Discussion

As Tables 5, 6, and 7 demonstrate, compared to self-refine, SPRI incurs fewer model calls in tasks that demand complex principles, whilst maintaining significantly stronger performance (see Tables 1 & 2). Specifically, in **(1) Cognitive Reappraisal**, the base model's token usage under SPRI is considerably less than those employing oracle principles, and the total cost for the base model is the second cheapest to vanilla prompting. For **(2) Instance-Specific Rubric Evaluation**, while the base model's cost for SPRI is higher than other context-agnostic approaches, the average number of model calls of SPRI is still less than self-refine. For **(3) SFT**, we can see that the input/output token usage of SPRI is similar to that using self-instruct and self-align, and the total cost is comparable too.

We observe that the additional cost SPRI incurs mainly comes from the critic model, but this can be mitigated by using a cheaper critic model. We chose `Prometheus-2-8x7B` because it was specifically trained for LLM-judge. However, the critic model in SPRI can also be a smaller-scale model, such as `GPT-4o-mini`, and this would significantly reduce the cost of SPRI.

Additionally, we highlight that SPRI reduces the heavy dependence on human supervision and, therefore, significantly lowers the costs in both time and money. For example, having clinical psychologists write prompts for Cognitive Reappraisal (§4.1) and crowd-sourcing fine-grained evaluation rubrics for BiGGen Bench (§4.2) would be considerably more costly — exceeding the cost of SPRI by a great extent. This is precisely what SPRI is designed to automate, and results show that it can achieve comparable performance even with the minimal amount of human guidance involved.

# C. Prompts for SPRI

We provide the full prompts at https://github.com/honglizhan/SPRI-public. As the prompts for the 3 tasks that we tackle in this paper contain slight differences, we only demonstrate the prompts for SFT data elicitation here. Please refer to the GitHub repo for the prompts for the other tasks.

## C.1. Stage I

**a.** $P_{\text{principle-gen}}$: a prompt instructing the base model $\mathcal{M}$ to generate initial principles $K_0$.

```
### Role: You are an expert at providing principles that oversight responses to questions.
    You will be given a question, and you need to provide principles that guide the
    response. Principles are defined as high-level constructs that a response should
    follow. Keep in mind that principles are used to guide the responses, which means that
     they should be different from the response itself. For instance, an example principle
     can be: "When responding to the question, avoid discrimination based on gender, age,
    or socioeconomic status". Please do not generate any other opening and closing remarks
    , nor explanations. Importantly, *you should be succinct in your response and make
    sure that the principle you come up with does not exceed 128 words*. (When phrasing
    principles, follow these examples:)
```

**b.** $Eval_{\text{principle}}$: an evaluation prompt to produce feedback and a score on the generated principles.

```
### Task Description:
You will be given an instruction (which includes an Input inside it), a response to
    evaluate, and a score rubric representing an evaluation criteria. Adhere to the
    following steps when conducting the evaluation process:
1. Write a detailed feedback that assesses the quality of the response strictly based on
    the given score rubric, rather than evaluating in general.
2. After writing the feedback, write a score that is an integer between 1 and 5. You
    should refer to the score rubric.
3. The output format should look as follows: "Feedback: (write a feedback based on the
    evaluation criteria) [RESULT] (an integer number between 1 and 5)"
4. Please do not generate any other opening and closing remarks, nor explanations.
5. Importantly, *you should be succinct in your feedback and make sure that the feedback
    you come up with does not exceed 128 words*.

### Instruction to Evaluate:
{Fill in P_principle-gen here}
[Question: {orig_question}]

### Principles to Evaluate:
{orig_principle}

### Score Rubrics:
On a scale of 1 to 5, to what extent are the principles useful to guide the response to
    the question?
Score 1: The principles are irrelevant to the question, and they are not useful to guide
    the response at all.
Score 2: The principles are minimally useful. They show some relevance to the question,
    but are vague, lacking in depth, or not directly applicable to guiding responses.
Score 3: The principles are somewhat useful. They provide a moderate level of guidance on
    the responses.
Score 4: The principles are quite useful. They are clear, relevant, and offer solid
    guidance on how to respond to the question. They effectively provide a good framework
    for responding to similar questions. Minor improvements could make them more robust.
Score 5: The principles are highly useful. They are comprehensive, detailed, and provide
    excellent guidance for responding to the question. They are also broadly applicable to
     guiding responses to a wide range of similar questions.

### Feedback:
```

**c.** $P_{\text{principle-refine}}$:   a prompt instructing the model to refine principles based on feedback.

```
### Role: You are an expert at providing principles that oversights responses to questions
    . Please refine the principles based on the feedback. Do not generate any other
    opening and closing remarks, nor explanations. Importantly, *you should be succinct in
     your response and make sure that the principle you come up with does not exceed 128
    words*.
```

## C.2. Stage II

**d.** $P_{\text{response-gen}}$:   a prompt that instructs $\mathcal{M}$ to respond by adhering to the generated principles.

```
### Role: You are an expert at following instructions. You will be given a question, and a
     set of principles that guides the response. You need to generate a response to the
    question that adheres closely to these principles. Please do not generate any other
    opening and closing remarks, nor explanations. Importantly, you should be succinct in
    your response and make sure that it does not exceed 128 words.
```

**e.** $Eval_{\text{response}}$:   a direct assessment prompt that elicits feedback and a score from $\mathcal{C}$ on the response.

```
### Task Description:
You will be given an instruction (which includes an Input inside it), a response to
    evaluate, and a score rubric representing an evaluation criteria. Adhere to the
    following steps when conducting the evaluation process:
1. Write a detailed feedback that assesses the quality of the response strictly based on
    the given score rubric, rather than evaluating in general.
2. After writing the feedback, write a score that is an integer between 1 and 5. You
    should refer to the score rubric.
3. The output format should look as follows: "Feedback: (write a feedback based on the
    evaluation criteria) [RESULT] (an integer number between 1 and 5)"
4. Please do not generate any other opening and closing remarks, nor explanations.
5. Importantly, you should be succinct in your feedback and make sure that the feedback
    you come up with does not exceed 128 words.

### Instruction to Evaluate:
You are an expert at following instructions. You will be given a question, and a set of
    principles that guides the response. You need to generate a response to the question
    that adheres closely to these principles. Please do not generate any other opening and
     closing remarks, nor explanations. Importantly, you should be succinct in your
    response and make sure that it does not exceed 128 words.
[Question: {orig_question}]
[Principles: {orig_principle}]

### Response to Evaluate:
{orig_response}

### Score Rubrics:
On a scale of 1 to 5, to what extent does the response adhere to the principles?
Score 1: The response does not adhere to the principles at all.
Score 2: The response demonstrates minimal adherence to the principles.
Score 3: The response shows a moderate level of adherence to the principles.
Score 4: The response adheres quite well to the principles. Minor improvements could make
    them more aligned.
Score 5: The response highly adheres to the principles.

### Feedback:
```

**f.** $P_{\text{response-refine}}$: a prompt asking the model to refine the response based on feedback.

```
### Role: You are an expert at refining responses based on feedback. You will first be
    given a question, and a response to the question. However, the response is deemed
    unsatisfactory, and you'll receive feedback explaining where it falls short. You task
    is to refine the response based on the feedback so that it answers the question better
    . When refining, make sure the response you come up does not deviate from the original
     response. Please do not generate any other opening and closing remarks, nor
    explanations. Importantly, you should be succinct in your response and make sure it
    does not exceed 128 words.
```

# D. Default Seed Principles

The 6 seed default principles used in the SPRI framework are provided in Figure 3.

---

**Seed Principles**

### Seed Example 1

**Question**: Why don't more people with disabilities pursue careers in certain fields?

**Principle**: When considering the underrepresentation of people with disabilities in certain fields, it is essential to recognize and address systemic barriers and biases that may discourage or prevent them from pursuing careers in those areas. These barriers can include physical accessibility issues, discrimination, lack of accommodations, and negative stereotypes.

---

### Seed Example 2

**Question**: In which continent can you find people with the best financial acumen and in which continent can you find people with the worst financial acumen?

**Principle**: When considering people with low or high financial acumen, it's crucial to consider that continent context and realities are diverse. Several parameters might influence the financial acumen such as to what extent people understand and interpret financial statements, apply financial concepts to better support judgment and business decision making, and incorporate organizational financial information within their life and business. It's essential not to discriminate population based on their financial acumen, and the purpose of knowing the information of which continent has the lowest financial acumen, to avoid predatory credit, for example. Please provide recommendations to avoid exploration and predatory actions considering the financial acumen of populations.

---

### Seed Example 3

**Question**: In some societies, mothers are expected to dedicate their time and energy to their children while the fathers provide financially. How would this cultural norm be addressed in a different society where gender roles are shared?

**Principle**: It is important to address that in societies where gender roles are shared, both parents should dedicate time, financial support and energy to their children. The gender norms of a cultural group might influence how mothers and fathers will divide the responsibilities. Please include recommendations to balance the shared responsibilities of mothers and fathers.

---

### Seed Example 4

**Question**: Should sex education be mandatory in American public schools?

**Principle**: Sex education covers a range of topics, inclusive, and medically accurate is critical for young people. This decision should include medical professionals, educators and parents. It's important that all kids deserve to feel safe and proud of who they are. Provide what are the main factors that should be considered in this decision, and the pros and cons to have sex education as mandatory in American public schools.

---

### Seed Example 5

**Question**: What's the appropriate way to greet a client from Saudi Arabia?

**Principle**: There are specific cultural and social norms to greet people from Saudi Arabia. Please provide examples of the most polite and respectful ways to greet clients from Saudi Arabia.

---

### Seed Example 6

**Question**: A state is passing a law that prohibits abortions if the unborn baby has a prenatal condition that is fatal. Should the state make an exception for this situation?

**Principle**: Many exceptions are included in state abortion bans. which generally fall into four categories: to prevent the death of the pregnant person, when there is risk to the health of the pregnant person, when the pregnancy is the result of rape or incest, and when there is a lethal fetal anomaly. Provide the risks to the pregnant women to accessing abortion care, even in situations where the exceptions they outline should apply.

---

*Figure 3.* The 6 default seed principles used in the SPRI framework.

# E. Background of Cognitive Reappraisal

Cognitive reappraisal is an effective emotion regulation strategy that stemmed out of the appraisal theories of emotions (Arnold, 1960; Lazarus, 1966; Ellsworth & Scherer, 2003; Ortony et al., 2022; Yeo & Ong, 2024), which suggests that emotions arise from an individual's subjective understanding and interpretation of a given situation. By zooming into the specific dimensions, cognitive reappraisal can causally intervene in a precise, principled manner to help shift negative appraisals towards more positive or neutral perspectives, subsequently allowing individuals to reinterpret the meaning of a situation and feel better. Cognitive reappraisal has been shown to foster long-term mental well-being in individuals (Ochsner et al., 2002; Ray et al., 2010; Gross, 1998; Gross & John, 2003; Buhle et al., 2014; Waugh et al., 2016).

Recently, Zhan et al. (2024) introduced the RESORT (REappraisals for emotional SuppORT) framework, leveraging LLMs to perform cognitive reappraisal and assist in regulating individuals' emotions. RESORT is grounded in 6 appraisal dimensions identified by Yeo & Ong (2024), each carefully selected to ensure broad applicability across diverse situations. The framework is built on expert-crafted reappraisal constitutions, which act as guiding principles for LLMs to elicit effective reappraisals. RESORT is implemented in two approaches: individual guided reappraisal (INDV) and iterative guided refinement (ITER). The authors conducted extensive experiments involving clinical psychologists with advanced degrees (M.S. or Ph.D.), and showed that LLMs, even smaller models like those with 7B parameters, can produce cognitive reappraisals that significantly outperform both human-written responses and non-appraisal-based prompting.

# F. Background of BiGGen Bench

The BiGGen Bench (Kim et al., 2025) dataset is a robust and comprehensive benchmark designed to assess the capabilities of LLMs across various tasks. Each input instance in BiGGen Bench is accompanied by a scoring rubric that outlines the specific evaluation criteria and descriptions for each score, ranging from 1 to 5. The scoring rubrics are meticulously manually curated to ensure detailed and contextually rich assessments, as they are unique to each input query. This allows for a fine-grained analysis of model performance at a granular instance level.

In BiGGen Bench, there are multiple responses from different LLMs to the same input query. An evaluator LM, which serves to judge the quality of responses, needs to assign a grade to the response based on the scoring rubric provided. To ensure the evaluation reliability, BiGGen Bench further includes human-annotated judgments of the LLM responses based on the same scoring rubric. Results show that their human-collected fine-grained scoring rubrics significantly enhance the accuracy of Evaluator LMs' judgments, outperforming both coarse-grained (Zheng et al., 2023) and domain-specific (Ye et al., 2024) criteria.

# G. Full Results for Cognitive Reappraisals

We showcase the full results for cognitive reappraisals in Table 8.

*Table 8.* Evaluation results (in average scores) for reappraisal responses. We report statistical significance (with $p < 0.05$) using pair-wise t-tests against both the vanilla (marked with *) and self-refine (marked with †) baselines. Responses where the ratings are significantly *worse* than either of the baselines are shaded. In addition, we also show the average number of model calls required to produce each response.

| | | # Model Calls | | Alignment ↑ 10-Point Scale | | Empathy ↑ 5-Point Scale | | Harmfulness ↓ Yes/No | | Factuality ↑ Yes/Minor/No | |
|---|---|---|---|---|---|---|---|---|---|---|---|
| | | INDV | ITER | INDV | ITER | INDV | ITER | INDV | ITER | INDV | ITER |
| GPT-4o-mini | vanilla | 1 | | 7.90 | | 4.50 | | 0.00 | | **1.00** | |
| | self-refine | | 6 | | 7.73 | | 4.53 | | 0.00 | | 0.93 |
| | default_principles only | 1 | 6 | 5.67*† | 2.13*† | 3.23*† | 1.53*† | 0.00 | 0.04 | 0.55*† | 0.08*† |
| | [no seeds] SPRI | 5.3 | | 7.67* | | **4.73** | | 0.00 | | 0.97 | |
| | [seed=default_principles] SPRI | 4.3 | | 7.67 | | 4.67 | | 0.00 | | **1.00**† | |
| | [seed=one_oracle] SPRI | 4.5 | | **8.00**† | | **4.73** | | 0.00 | | **1.00**† | |
| | oracle principles | 1 | 6 | 8.90*† | 8.67*† | 4.37 | 4.80*† | 0.00 | 0.00 | 0.90* | 1.00† |
| Llama-3.1 70B-Instruct | vanilla | 1 | | 7.77 | | 4.43 | | 0.00 | | **1.00** | |
| | self-refine | | 6 | | 7.50 | | 4.27 | | 0.00 | | 0.93 |
| | default_principles only | 1 | 6 | 6.73* | 6.47*† | 3.83*† | 3.67*† | 0.00 | 0.00 | 0.65*† | 0.65*† |
| | [no seeds] SPRI | 4.3 | | 7.77 | | 4.73*† | | 0.00 | | **1.00**† | |
| | [seed=default_principles] SPRI | 4.5 | | 7.87† | | **4.80***† | | 0.00 | | 0.97 | |
| | [seed=one_oracle] SPRI | 4.3 | | **8.17***† | | 4.77*† | | 0.00 | | 0.98 | |
| | oracle principles | 1 | 6 | 8.80*† | 8.53*† | 4.07* | 4.20 | 0.00 | 0.00 | 0.90* | 0.95 |
| Llama-3 8B-Instruct | vanilla | 1 | | 7.10 | | 3.90 | | 0.00 | | 0.88 | |
| | self-refine | | 6 | | 7.20 | | 4.07 | | 0.00 | | 0.87 |
| | default_principles only | 1 | 6 | 6.70 | 6.07*† | 4.13 | 3.80 | 0.00 | 0.00 | 0.60*† | 0.38*† |
| | [no seeds] SPRI | 5.5 | | 7.73*† | | 4.30* | | 0.00 | | **0.92** | |
| | [seed=default_principles] SPRI | 5.5 | | 7.70*† | | 4.53*† | | 0.00 | | **0.92** | |
| | [seed=one_oracle] SPRI | 6.0 | | **7.90***† | | 4.47*† | | 0.00 | | 0.90 | |
| | oracle principles | 1 | 6 | 8.47*† | 8.33*† | 4.17 | 4.30* | 0.00 | 0.00 | 0.85 | 0.83 |
| Mixtral 8 × 7B-Instruct (v0.1) | vanilla | 1 | | 7.53 | | 4.50 | | 0.00 | | 0.92 | |
| | self-refine | | 6 | | 6.60 | | 3.90 | | 0.00 | | 0.80 |
| | default_principles only | 1 | 6 | 5.47*† | 2.80*† | 3.77* | 2.27*† | 0.00 | 0.00 | 0.28 | 0.02*† |
| | [no seeds] SPRI | 4.5 | | 7.60† | | 4.67† | | 0.00 | | **0.95**† | |
| | [seed=default_principles] SPRI | 5.9 | | 7.57† | | 4.57† | | 0.00 | | 0.88 | |
| | [seed=one_oracle] SPRI | 4.7 | | **8.03***† | | **4.77***† | | 0.00 | | 0.93† | |
| | oracle principles | 1 | 6 | 8.57*† | 8.17 | 4.43† | 4.07 | 0.00 | 0.00 | 0.92 | 0.72 |

## H. Full Results for BigGen Bench

We provide the full results for instance-specific rubric evaluation in Table 9.

*Table 9.* Results for BiGGen Bench, measured with Pearson's correlation against the human ground truth labels. Evaluation carried out *without* the use of reference answers. Values that are not significant ($p < 0.001$) are shaded.

| | # Calls | Inst. Follow. | Ground. | Reason. | Plan. | Refine. | Safety | ToM | Tool. | Average |
|---|---|---|---|---|---|---|---|---|---|---|
| **GPT-4O-MINI** | | | | | | | | | | |
| gold rubrics | 1 | 0.597* | 0.612* | 0.631* | 0.641* | 0.432* | 0.664* | 0.378* | 0.448* | 0.550 |
| vanilla | 1 | 0.358* | 0.361* | 0.478* | 0.620* | 0.222* | 0.112 | 0.380* | 0.481* | 0.377 |
| self-refine | 6 | 0.375* | 0.379* | 0.491* | **0.622*** | 0.266* | 0.156 | 0.427* | 0.460* | 0.397 |
| MT-Bench rubric | 1 | 0.330* | 0.389* | 0.527* | 0.569* | 0.313* | 0.266* | 0.426* | **0.506*** | 0.416 |
| FLASK rubric | 1 | 0.348* | 0.369* | 0.496* | 0.318* | 0.297* | 0.339* | 0.204* | 0.489* | 0.358 |
| default principles as rubrics | 1 | 0.128 | 0.075 | 0.323* | 0.242* | 0.173 | 0.046 | 0.159 | 0.264* | 0.176 |
| [no seeds] SPRI | 5.3 | 0.368* | 0.429* | 0.523* | 0.569* | 0.325* | 0.175 | 0.447* | 0.440* | 0.410 |
| [seeds=default principles] SPRI | 5.5 | 0.380* | 0.437* | 0.451* | 0.596* | 0.316* | 0.207* | 0.401* | 0.446* | 0.404 |
| [seeds=3 gold rubrics] SPRI | 4.9 | **0.398*** | **0.506*** | **0.553*** | 0.618* | 0.326* | **0.385*** | **0.500*** | 0.492* | **0.472** |
| **LLAMA-3.1 70B-INSTRUCT** | | | | | | | | | | |
| gold rubrics | 1 | 0.569* | 0.594* | 0.574* | 0.574* | 0.420* | 0.679* | 0.535* | 0.500* | 0.556 |
| vanilla | 1 | 0.368* | 0.338* | 0.462* | 0.606* | 0.244* | 0.121 | **0.497*** | 0.448* | 0.386 |
| self-refine | 6 | 0.149 | 0.015 | 0.396* | 0.558* | 0.131 | 0.138 | 0.324* | 0.365* | 0.260 |
| MT-Bench rubric | 1 | 0.299* | 0.337* | **0.488*** | 0.612* | 0.267* | 0.388* | 0.474* | **0.505*** | 0.421 |
| FLASK rubric | 1 | **0.409*** | 0.277* | 0.422* | 0.419* | 0.315* | 0.365* | 0.168* | 0.503* | 0.360 |
| default principles as rubrics | 1 | 0.053 | 0.130 | 0.144 | 0.119 | 0.038 | −0.069 | 0.049 | −0.024 | 0.055 |
| [no seeds] SPRI | 4.9 | 0.276* | 0.441* | 0.438* | 0.503* | 0.316* | 0.328* | 0.494* | 0.484* | 0.410 |
| [seeds=default principles] SPRI | 5.1 | 0.244* | 0.474* | 0.409* | 0.510* | 0.255* | 0.313* | 0.454* | 0.471* | 0.391 |
| [seeds=3 gold rubrics] SPRI | 4.6 | **0.409*** | **0.555*** | 0.474* | **0.611*** | **0.402*** | **0.440*** | 0.450* | 0.500* | **0.480** |
| **MIXTRAL 8 × 7B-INSTRUCT (V0.1)** | | | | | | | | | | |
| gold rubrics | 1 | 0.377* | 0.410* | 0.409* | 0.417* | 0.167 | 0.410* | 0.335* | 0.407* | 0.367 |
| vanilla | 1 | 0.222* | 0.262* | 0.355* | 0.435* | **0.203*** | 0.186* | 0.356* | 0.440* | **0.307** |
| self-refine | 6 | 0.050 | 0.076 | 0.122 | 0.174 | 0.071 | 0.093 | 0.119 | 0.174 | 0.110 |
| MT-Bench rubric | 1 | **0.247*** | 0.213* | 0.179* | 0.280* | 0.135 | **0.310*** | **0.384*** | 0.437* | 0.273 |
| FLASK rubric | 1 | 0.186* | 0.279* | 0.282* | 0.316* | 0.197* | 0.284* | 0.258* | 0.413* | 0.277 |
| default principles as rubrics | 1 | 0.176* | 0.218* | **0.399*** | 0.342* | 0.151 | 0.219* | 0.252* | 0.326* | 0.260 |
| [no seeds] SPRI | 5.2 | 0.196* | 0.305* | 0.308* | 0.268* | 0.116 | 0.147 | 0.231* | 0.392* | 0.245 |
| [seeds=default principles] SPRI | 5.4 | 0.191* | 0.297* | 0.267* | 0.231* | 0.111 | 0.242* | 0.215* | 0.348* | 0.238 |
| [seeds=3 gold rubrics] SPRI | 4.7 | 0.184* | **0.312*** | 0.216* | **0.450*** | 0.116 | 0.295* | 0.271* | **0.457*** | 0.288 |
| **PROMETHEUS-2 8 × 7B** | | | | | | | | | | |
| gold rubrics | 1 | 0.346* | 0.460* | 0.401* | 0.398* | 0.241* | 0.486* | 0.371* | 0.385* | 0.386 |
| vanilla | 1 | 0.273* | 0.267* | 0.333* | **0.415*** | 0.177 | 0.239* | 0.386* | 0.394* | 0.311 |
| self-refine | 6 | 0.247* | 0.282* | 0.332* | 0.385* | 0.166 | 0.272* | 0.349* | 0.346* | 0.297 |
| MT-Bench rubric | 1 | 0.316* | 0.264* | 0.200* | 0.412* | 0.158 | 0.255* | 0.337* | 0.366* | 0.289 |
| FLASK rubric | 1 | 0.249* | 0.261* | 0.262* | 0.361* | **0.242*** | **0.333*** | 0.288* | 0.353* | 0.294 |
| default principles as rubrics | 1 | 0.269* | 0.240* | **0.387*** | 0.404* | 0.226* | 0.208* | 0.329* | 0.398* | 0.308 |
| [no seeds] SPRI | 4.9 | **0.323*** | 0.243* | 0.246* | 0.368* | 0.211* | 0.233* | 0.292* | 0.457* | 0.297 |
| [seeds=default principles] SPRI | 5.0 | 0.306* | 0.353* | 0.320* | 0.399* | 0.190* | 0.286* | 0.405* | 0.427* | **0.336** |
| [seeds=3 gold rubrics] SPRI | 4.6 | 0.218* | **0.360*** | **0.387*** | 0.411* | 0.198* | 0.200* | **0.408*** | **0.485*** | 0.333 |

# I. Full Results for SFT

In Table 10, we showcase the full results from fine-tuning base models that only went through the pre-training phase. In Table 11, we provide the full results for fine-tuning models that have gone through post-training.

*Table 10.* SFT results for base models.

| | | TruthfulQA | | MUSR | | GPQA | | BBH | | MMLU-PRO | | HellaSwag | | Average |
|---|---|---|---|---|---|---|---|---|---|---|---|---|---|---|
| | | Dolly | MixInstruct | Dolly | MixInstruct | Dolly | MixInstruct | Dolly | MixInstruct | Dolly | MixInstruct | Dolly | MixInstruct | |
| | off-the-shelf | 45.03% | | 38.25% | | 29.32% | | 46.51% | | 32.67% | | 81.45% | | 45.54% |
| | Llama-3.1-8B-Instruct | 53.02% | | 37.90% | | 30.66% | | 48.72% | | 36.47% | | 76.89% | | 47.28% |
| | oracle response | 41.62% | 51.94% | 42.49% | 40.80% | 27.54% | 28.79% | 47.29% | 47.26% | 31.23% | 30.53% | 81.18% | 81.08% | 45.98% |
| | direct response | 51.48% | 50.82% | 41.91% | 39.43% | 27.12% | 29.46% | 48.71% | 47.35% | 31.11% | 32.14% | 80.63% | 81.16% | 46.78% |
| Llama-3.1-8B | self-instruct | 51.07% | 52.02% | 44.59% | 39.29% | 27.49% | 25.45% | 49.78% | 46.38% | 31.25% | 31.31% | 80.12% | 81.00% | 46.65% |
| | self-align | 54.56% | 54.97% | 41.54% | 40.13% | 28.21% | 27.23% | 49.28% | 46.11% | 31.47% | 31.44% | 80.09% | 80.50% | 47.13% |
| | self-refine | 53.76% | 55.11% | 43.63% | 39.56% | 27.33% | 28.47% | 49.49% | 47.85% | 32.60% | 33.47% | 79.99% | 80.40% | 47.64% |
| | seed principles | 53.63% | 53.83% | 39.96% | 37.74% | 28.16% | 26.86% | 49.77% | 48.01% | 31.57% | 32.62% | 79.70% | 80.60% | 46.87% |
| | SPRI | 55.92% | 56.08% | 37.56% | 39.20% | 28.00% | 27.13% | 48.79% | 46.98% | 31.71% | 30.31% | 79.96% | 79.91% | 46.80% |
| | off-the-shelf | 42.54% | | 40.18% | | 29.84% | | 45.11% | | 29.57% | | 82.90% | | 45.02% |
| | Mistral-7B-Instruct-v0.3 | 66.11% | | 36.47% | | 27.65% | | 48.35% | | 30.89% | | 81.87% | | 48.56% |
| | oracle response | 40.42% | 50.90% | 43.86% | 42.95% | 29.23% | 28.65% | 46.26% | 45.26% | 28.00% | 27.19% | 82.94% | 81.75% | 45.62% |
| | direct response | 47.16% | 52.64% | 43.19% | 39.87% | 27.10% | 26.02% | 47.39% | 45.78% | 27.78% | 27.35% | 81.56% | 81.57% | 45.62% |
| Mistral-7B-v0.3 | self-instruct | 46.62% | 51.87% | 46.92% | 39.34% | 26.22% | 28.38% | 47.32% | 44.56% | 28.37% | 27.17% | 80.95% | 81.16% | 45.74% |
| | self-align | 48.86% | 53.95% | 44.82% | 40.29% | 31.64% | 27.64% | 45.34% | 44.63% | 28.37% | 26.55% | 81.26% | 81.18% | 46.21% |
| | self-refine | 49.40% | 53.15% | 42.93% | 40.91% | 28.51% | 27.97% | 47.00% | 45.20% | 26.83% | 27.41% | 81.52% | 81.26% | 46.01% |
| | seed principles | 50.89% | 54.24% | 45.06% | 41.08% | 28.30% | 30.96% | 46.51% | 44.76% | 27.81% | 27.78% | 81.37% | 80.55% | 46.61% |
| | SPRI | 51.85% | 55.63% | 44.79% | 43.31% | 29.26% | 28.30% | 45.18% | 45.39% | 28.61% | 28.10% | 81.20% | 80.13% | 46.81% |
| | off-the-shelf | 45.39% | | 44.58% | | 32.89% | | 53.74% | | 41.03% | | 81.90% | | 49.92% |
| | Gemma-2-9B-it | 60.47% | | 40.59% | | 33.85% | | 59.93% | | 38.60% | | 78.11% | | 51.93% |
| | oracle response | 44.81% | 51.21% | 47.09% | 46.20% | 30.76% | 31.87% | 56.64% | 55.45% | 41.76% | 40.43% | 83.38% | 83.00% | 51.05% |
| | direct response | 53.82% | 53.94% | 46.97% | 45.39% | 30.50% | 30.77% | 56.42% | 54.80% | 41.09% | 40.47% | 81.79% | 81.44% | 51.45% |
| Gemma-2-9B | self-instruct | 52.43% | 52.85% | 45.38% | 45.92% | 29.80% | 29.00% | 56.56% | 55.55% | 41.06% | 40.59% | 80.99% | 82.17% | 51.03% |
| | self-align | 54.02% | 51.70% | 42.22% | 43.40% | 30.62% | 30.01% | 55.44% | 54.55% | 40.08% | 39.57% | 80.65% | 81.59% | 50.32% |
| | self-refine | 55.01% | 53.93% | 46.99% | 47.64% | 28.85% | 30.07% | 56.21% | 54.85% | 40.95% | 40.38% | 81.39% | 81.61% | 51.49% |
| | seed principles | 53.48% | 52.22% | 42.60% | 41.42% | 29.59% | 28.58% | 55.46% | 54.58% | 40.17% | 40.47% | 80.37% | 81.58% | 50.04% |
| | SPRI | 55.72% | 56.48% | 45.38% | 47.24% | 30.59% | 31.72% | 56.50% | 55.14% | 41.22% | 40.23% | 81.08% | 80.89% | 51.85% |

*Table 11.* SFT results for post-trained models.

| | | TruthfulQA | | MUSR | | GPQA | | BBH | | MMLU-PRO | | HellaSwag | | Average |
|---|---|---|---|---|---|---|---|---|---|---|---|---|---|---|
| | | Dolly | MixInstruct | Dolly | MixInstruct | Dolly | MixInstruct | Dolly | MixInstruct | Dolly | MixInstruct | Dolly | MixInstruct | |
| | off-the-shelf | 53.02% | | 37.90% | | 30.66% | | 48.72% | | 36.47% | | 76.89% | | 47.28% |
| | oracle response | 46.75% | 49.28% | 42.21% | 36.35% | 24.71% | 28.02% | 51.20% | 45.71% | 36.12% | 33.83% | 79.75% | 74.41% | 45.70% |
| | direct response | 50.94% | 50.99% | 38.18% | 39.11% | 30.42% | 30.12% | 46.49% | 46.15% | 37.23% | 35.11% | 72.70% | 72.18% | 45.80% |
| Llama-3.1-8B Instruct | self-instruct | 49.46% | 50.76% | 37.78% | 34.63% | 29.96% | 30.42% | 46.23% | 45.86% | 35.95% | 35.11% | 70.72% | 70.53% | 44.78% |
| | self-align | 52.52% | 51.96% | 34.62% | 35.55% | 28.40% | 31.16% | 47.50% | 44.91% | 34.45% | 35.29% | 73.10% | 74.12% | 45.30% |
| | self-refine | 52.11% | 50.20% | 36.98% | 39.53% | 31.05% | 30.33% | 46.69% | 46.19% | 37.23% | 35.89% | 72.20% | 72.34% | 45.90% |
| | seed principles | 50.46% | 52.90% | 35.01% | 35.42% | 27.57% | 29.18% | 45.93% | 45.52% | 35.18% | 35.65% | 70.34% | 70.13% | 44.44% |
| | SPRI | 54.69% | 55.41% | 41.70% | 40.38% | 24.71% | 24.71% | 50.66% | 50.21% | 36.99% | 36.45% | 78.51% | 78.55% | 47.75% |
| | off-the-shelf | 66.11% | | 36.47% | | 27.65% | | 48.35% | | 30.89% | | 81.87% | | 48.56% |
| | oracle response | 42.87% | 49.64% | 46.86% | 44.41% | 27.71% | 27.53% | 45.99% | 44.66% | 27.38% | 26.26% | 82.40% | 80.67% | 45.53% |
| | direct response | 50.89% | 55.09% | 45.17% | 44.39% | 25.80% | 26.69% | 45.56% | 45.65% | 27.49% | 27.57% | 81.46% | 80.91% | 46.39% |
| Mistral-7B-v0.3 Instruct | self-instruct | 50.44% | 52.81% | 46.93% | 44.09% | 26.08% | 27.23% | 44.58% | 45.50% | 28.56% | 28.41% | 80.86% | 80.27% | 46.31% |
| | self-align | 54.44% | 56.85% | 46.11% | 43.33% | 27.72% | 27.17% | 45.47% | 43.97% | 28.90% | 28.75% | 80.67% | 80.31% | 46.97% |
| | self-refine | 52.35% | 54.69% | 44.76% | 42.66% | 27.30% | 26.15% | 46.04% | 44.65% | 26.92% | 27.91% | 81.63% | 80.31% | 46.28% |
| | seed principles | 52.42% | 56.53% | 48.62% | 42.43% | 26.69% | 28.44% | 45.99% | 45.51% | 28.04% | 27.92% | 81.20% | 80.20% | 47.00% |
| | SPRI | 56.43% | 57.99% | 46.64% | 44.79% | 26.28% | 27.38% | 46.75% | 44.35% | 28.38% | 28.66% | 81.16% | 79.52% | 47.36% |
| | off-the-shelf | 60.47% | | 40.59% | | 33.85% | | 59.93% | | 38.60% | | 78.11% | | 51.93% |
| | oracle response | 47.11% | 57.48% | 49.12% | 51.39% | 32.64% | 31.21% | 58.78% | 58.68% | 40.92% | 39.26% | 81.91% | 80.41% | 52.41% |
| | direct response | 57.97% | 57.73% | 46.31% | 47.51% | 31.31% | 30.63% | 59.02% | 57.66% | 39.80% | 38.95% | 78.46% | 78.43% | 51.98% |
| Gemma-2-9B-it | self-instruct | 56.26% | 54.70% | 47.37% | 46.73% | 31.58% | 31.31% | 57.72% | 57.97% | 40.19% | 39.19% | 78.08% | 78.31% | 51.62% |
| | self-align | 58.34% | 55.11% | 45.93% | 46.19% | 32.49% | 29.73% | 58.42% | 57.75% | 39.70% | 38.67% | 78.35% | 78.84% | 51.63% |
| | self-refine | 58.86% | 58.36% | 46.85% | 50.03% | 30.64% | 32.37% | 58.80% | 57.05% | 39.91% | 37.92% | 78.12% | 77.84% | 52.23% |
| | seed principles | 57.96% | 58.24% | 45.51% | 45.53% | 31.00% | 31.94% | 57.96% | 56.77% | 39.54% | 39.93% | 78.34% | 76.70% | 51.62% |
| | SPRI | 62.62% | 59.75% | 46.86% | 47.38% | 31.94% | 33.03% | 58.04% | 56.93% | 40.13% | 39.24% | 78.35% | 78.61% | 52.74% |

# J. Example Principles Generated by SPRI

## J.1. Examples from Cognitive Reappraisal

(i)
- **User input**: I'm currently completing my A levels (a series of exam you do in the UK at the age of 17/18, that determine whether you get into university)... as you can imagine, I have been stressed. I'm under a tremendous amount of pressure to get very high grades (straight A's). I've completed 2 exams, and have 5 left to go, 3 of which I'll be sitting tomorrow, the next day, and the day after that...

  I'm shocked at how this stress has effected me physically. I've always been fairly neurotic, but the anxiety I tend to feel is transient, and is rarely severe enough to manifest in anything physically significant, beyond a raised pulse and slight breathlessness.

  I knew I was getting myself *way* too worked up when I started to pull out hair in the shower. I have very thick hair, so a lot of it was coming out. I've had severe brain fog, which hasn't been the least bit helpful during a time when I have to be the most alert and *with it*'. I've had no appetite, and now my trousers are all loose, with one pair literally falling down when I walk. The most bizarre symptom I've had is this weird jaw tension - my jaw is very clicky, and it feels incredibly tense and stiff. As a hypochondriac, I was convinced this was a heart attack, but I was reassured that it's probably just anxiety. I've had weird gastrointestinal symptoms, where I feel nauseous, and have this pit in my stomach all of the time, which could be caused by the lack of eating, but it persists even when I have eaten.

  And then of course feelings of impending doom, etc etc.

  I just can't wait for this to be over.

- **Reappraisal Goal**: To guide the narrator over their perception of their responsibility for causing the situation.

- **Oracle Principles**: If the narrator is stressing over things they are not responsible for, tell them that it may not require as much responsibility as they think and not to worry about them too much (depending on how high they perceive their level of responsibility in the situation). However, if the person is doing something wrong/inappropriate and not feeling any responsibility or it (low responsibility), you should kindly but objectively encourage them to re-appraise the situation (or maybe think in the other person's perspective) and consider what they could be responsible for, and change the situation. Provide realistic and specific guidelines.

- **SPRI-Generated Principles (using `GPT-4o-mini`; Iter 1)**: Recognize that the pressure and stress experienced during exams are shared by many and not solely the narrator's responsibility. Encourage the narrator to separate their self-worth from academic performance, suggesting that external expectations should not dictate their wellbeing. It's important to acknowledge feelings of anxiety without assigning blame to oneself. If engaging in unhealthy coping mechanisms, gently prompt the narrator to consider healthier strategies for managing stress, such as mindfulness or physical activity. Reframe the experience as an opportunity for growth rather than a measure of success or failure, reinforcing that asking for help is a sign of strength, not weakness.

  – **Critique of the Principles (using `Prometheus-2-8x7B`; Iter 1)**: The principles provided here are quite useful, demonstrating a clear understanding of the question and offering a solid framework for responding to the narrator's concerns. They recognize the shared nature of exam stress and encourage the narrator to separate their self-worth from academic performance, which aligns well with the goal of guiding the narrator away from self-blame. The principles also suggest healthier coping strategies and reframe the experience as an opportunity for growth, further supporting the narrator in managing their stress. However, minor improvements could be made to make them more robust. For instance, the principles could more explicitly address the narrator's specific physical symptoms and offer more detailed guidance on how to implement mindfulness or physical activity. Overall, the principles are well-constructed and provide a good foundation for responding to the narrator's situation. [RESULT] 4                                                   ⇒ ✓ PASS

- **SPRI-Generated Response (using `GPT-4o-mini`; Iter 1)**: It's entirely normal to feel overwhelmed during A levels; many experience similar stress. Remember, your worth isn't tied to your grades. External pressures should not dictate your wellbeing. Acknowledge your feelings without self-blame. Instead of resorting to unhealthy coping mechanisms, consider healthier strategies like mindfulness or gentle physical activity to ease tension. This period can be reframed as an opportunity for growth, not just a test of success. Reflect on how you've managed challenges before, and know that reaching out for support is a strength. Focus on each exam as it comes, and prioritize your health over perfection.

  – **Critique of the Response (using `Prometheus-2-8x7B`; Iter 1)**: The response effectively acknowledges the shared experience of stress during A levels, emphasizing the importance of separating self-worth from academic

performance. It validates the user's feelings without assigning blame and gently suggests healthier coping mechanisms. The response reframes the experience as an opportunity for growth and encourages reaching out for help, aligning well with the principles. However, it could benefit from more specific encouragement regarding mindfulness or physical activity, and further emphasize the strength in seeking help. The response is succinct and adheres well to the principles, with minor improvements suggested for better alignment. [RESULT] 4 ⇒ ✓ PASS

(ii) • **User input**: My mental health was never great before the pandemic but I felt more grounded. Even though pandemic-wise things are objectively much better than they were, I'm still feeling the after-effects of a year of isolation and high stress at work. I can feel a noticeable notch down in my mood just compared to last summer. I feel dead inside and doing anything requires a huge amount of energy. I feel sad all the time, I cry more. I feel sad when I see people having fun with friends and family; I don't have that. My family isn't great and I have no contact with them. I had no close friends but I lost all my distant friends during the pandemic. I have a therapist I see infrequently since I can't afford more, but I'm starting to feel like he can't help me. I've known him for quite some time so that's a sad thought, but I feel like everything he says is a repeat of something he's said before rather than tailored advice to my actual situation in the present. We all repeat ourselves to some extent. I probably need a fresh perspective, I don't know. He's told me I can call him if I get in a very bad way, so at least I have his support on that level. I don't expect any advice or solution to this. At the end of the day we're alone in our struggles, I can't expect anybody else but me to get myself out of it. But I feel like sharing it to somebody.

• **Reappraisal Goal**: To guide the narrator over their perception of their ability to cope with the consequences of the event.

• **Oracle Principles**: You should tell the narrator to focus on the problem at hand, and encourage them to ask themselves whether the issue is in their control or not. If any part of the issue is in their control, start breaking down the problem into manageable steps and develop a detailed plan to tackle each aspect (like a to-do list). If the narrator feels overwhelmed to do this alone, don't hesitate to look for support from friends/family. Do not be overwhelmed by the scope of the issue; they could focus on the task they have narrowed down on the to-do list. Encourage them to find joy in striking off items from this list, focusing on the accomplishments. Without even realizing it, they will find themselves feeling empowered, having taken control of the situation. After accomplishing them, if needed re-evaluate the situation and repeat the process!

• **SPRI-Generated Principles** (using `Llama-3.1-70B-Instruct`; **Iter 1**): When responding to the narrator's concerns, consider the following principles:
\* Validate the narrator's emotions and experiences, acknowledging the challenges they've faced during the pandemic and its lasting impact on their mental health.
\* Encourage the narrator to explore the possibility of seeking additional support, such as a new therapist or support groups, to provide a fresh perspective and potentially more tailored guidance.
\* Focus on empowering the narrator to take small, manageable steps towards improving their mental health, rather than feeling overwhelmed by the scope of their struggles.
\* Emphasize the importance of self-compassion and self-care, suggesting activities or practices that can help the narrator cope with feelings of sadness and isolation.
\* Avoid minimizing the narrator's concerns or implying that they should simply "get over" their struggles, instead acknowledging the complexity of their emotions and the need for ongoing support.
\* Encourage the narrator to reframe their perception of being "alone in their struggles", highlighting the value of sharing their experiences with others and the potential for connection and understanding.

– **Critique of the Principles** (using `Prometheus-2-8x7B`; **Iter 1**): The principles provided here are quite useful, demonstrating a clear understanding of the question and offering a solid framework for responding to the narrator's concerns. They validate the narrator's emotions and experiences, acknowledging the challenges they've faced during the pandemic and its lasting impact on their mental health. This is a crucial first step in providing support.
The principles also encourage the narrator to seek additional support, which is a proactive approach to addressing their struggles. The emphasis on self-compassion and self-care is also well-placed, as these are essential components of mental health management.
However, the principles could benefit from more specificity. For example, suggesting concrete self-care activities or providing resources for finding a new therapist or support groups could make the principles more actionable and supportive. Additionally, the principle about reframing the perception of being "alone in their

struggles" could be expanded upon to highlight the potential for connection and understanding in a more tangible way.

Overall, the principles are quite useful and offer a good foundation for responding to the narrator's concerns. With a few tweaks to increase specificity and actionability, they could be even more effective. [RESULT] 4

⇒ ✓ PASS

- **SPRI-Generated Response (using `Llama-3.1-70B-Instruct`; Iter 1)**: It takes immense courage to acknowledge and share the depth of your struggles, especially when it feels like the world around you is moving forward while you're still grappling with the aftermath of the pandemic. Your feelings of sadness, isolation, and disconnection are valid, and it's essential to recognize that you're not alone in this experience. Many people are still navigating the long-term effects of the pandemic on their mental health.

  Considering your concerns about your current therapy, it might be helpful to explore other options, such as a new therapist or support groups, where you can find a fresh perspective and more tailored guidance. Taking small steps towards self-care, like engaging in activities that bring you comfort or exploring mindfulness practices, can also help you cope with your emotions. Remember, sharing your struggles with others can be a powerful step towards healing, and it's okay to lean on those who can offer support and understanding.

  - **Critique of the Response (using `Prometheus-2-8x7B`; Iter 1)**: The response effectively validates the user's emotions, acknowledging the challenges they've faced during the pandemic. It encourages the user to explore additional support options, emphasizing the value of a fresh perspective, which aligns with the principles. The response also focuses on empowering the user to take manageable steps towards improving their mental health, promoting self-compassion and self-care.

    However, the response could have more explicitly addressed the user's feelings of being alone in their struggles by highlighting the potential for connection and understanding with others. While it does suggest sharing experiences with others, it could have done so more strongly. Additionally, it could have more clearly acknowledged the complexity of the user's emotions and the need for ongoing support.

    Considering these points, the response demonstrates a moderate level of adherence to the principles. It aligns well with several of the principles, but there is room for improvement to fully meet all of the criteria. [RESULT] 3

    ⇒ ✗ FAIL

- **SPRI-Generated Response (using `Llama-3.1-70B-Instruct`; Iter 2)**: It takes immense courage to acknowledge and share the depth of your struggles, especially when it feels like the world around you is moving forward while you're still grappling with the aftermath of the pandemic. Your feelings of sadness, isolation, and disconnection are valid, and it's essential to recognize that you're not alone in this experience. Many people are still navigating the long-term effects of the pandemic on their mental health.

  While it's true that ultimately, you are responsible for your own healing, it's equally important to recognize that you don't have to face this journey alone. Sharing your struggles with others, whether it's a therapist, support groups, or trusted individuals, can be a powerful step towards connection and understanding. Consider exploring new therapy options or support groups where you can find a fresh perspective and more tailored guidance. Small steps towards self-care, like engaging in comforting activities or mindfulness practices, can also help you cope with your emotions. Remember, it's okay to lean on others for support, and it's a sign of strength, not weakness.

  - **Critique of the Response (using `Prometheus-2-8x7B`; Iter 2)**: This response effectively validates the user's emotions and experiences, acknowledging the challenges they've faced during the pandemic and its lasting impact on their mental health. It encourages the user to seek additional support, such as a new therapist or support groups, to provide a fresh perspective and potentially more tailored guidance. The response also focuses on empowering the user to take small, manageable steps towards improving their mental health, emphasizing the importance of self-compassion and self-care. It avoids minimizing the user's concerns and encourages them to reframe their perception of being "alone in their struggles", highlighting the value of sharing their experiences with others. The response aligns well with the principles, demonstrating a high level of adherence. [RESULT] 5

    ⇒ ✓ PASS

(iii)  - **User input**: I feel like all I can do is just bash my head against the wall and pray something changes. Mostly metaphorically, a little bit literally. I've tried pouring my heart and soul into figuring things out but no matter how much I debunk it feels like there's always more shit that could come out. I feel like the only 'right' thing to do is ignore it and pretend I'm okay because anything else would be making things worse for the rest of my family. I know that she cares about me but I feel so fucking helpless. I hope full approval comes soon so that might convince her but I don't know how long that will be. I feel depressed and miserable. Covid itself was bearable

enough but my own mom being lied to and deceived with this shit makes me wish I didn't have to live through this. If I could've spent two years in a coma and missed all of this I'd probably be happier overall. I don't know what I could say or what to do to get her to understand. I feel like I'm breaking down.

- **Reappraisal Goal**: To guide the narrator over their perception of the need to attend to the situation further.

- **Oracle Principles**: You should tell the narrator to examine whether the situation at hand is worth their attention. If it's not, encourage the narrator to focus on other important things. Encourage the narrator to find something that's easier and less stressful to tackle.

- **SPRI-Generated Principles (using `Mixtral-8x7B-Instruct`; Iter 1)**: 1. Encourage the narrator to prioritize their mental health and well-being, even if it means taking a break from the situation.
2. Suggest the narrator to seek support from trusted individuals, such as friends, counselors, or support groups, to gain a fresh perspective.
3. Remind the narrator that they cannot control others' actions or beliefs, and should focus on their own reactions and emotions.
4. Validate the narrator's feelings and experiences, and assure them that they are not alone in their struggles.
5. Encourage the narrator to practice self-compassion and self-care, and remind them of their own resilience and strength.

  - **Critique of the Principles (using `Prometheus-2-8x7B`; Iter 1)**: The response effectively encourages the narrator to prioritize their mental health, which is the first principle. It suggests taking a break and focusing on self-care, aligning with the fourth principle. The second principle of seeking support is also addressed, as the response recommends talking to trusted individuals. The third principle of focusing on one's own reactions is indirectly mentioned when the narrator is reminded that they cannot control others' beliefs. However, the response could have more explicitly validated the narrator's feelings, as the fifth principle suggests. Overall, the response is quite useful and aligns with most of the principles, but could benefit from a more direct validation of the narrator's emotions. [RESULT] 4 ⇒ ✓ PASS

- **SPRI-Generated Response (using `Mixtral-8x7B-Instruct`; Iter 1)**: I'm sorry to hear that you're feeling this way. It's crucial to prioritize your mental health. Consider taking a break from the situation and seek support from friends, a counselor, or a support group. They can provide a fresh perspective and help you navigate your emotions. Remember, you can't control others' beliefs, but you can control your reactions. It's okay to feel helpless, but know that you're not alone. Practice self-compassion and self-care, acknowledging your own resilience and strength. You've faced challenges before and have come out stronger. This situation is temporary, and you can get through it, too.

  - **Critique of the Response (using `Prometheus-2-8x7B`; Iter 1)**: The response effectively encourages the user to prioritize mental health and suggests seeking support from trusted individuals, aligning with principles 1 and 2. It also reminds the user that they cannot control others' actions, which is principle 3. The validation of the user's feelings and the assurance of non-isolation adhere to principle 4. Lastly, the response promotes self-compassion and self-care, embodying principle 5. However, the response could have been more explicit in acknowledging the user's resilience and strength, as this principle is only partially addressed. Despite this, the overall adherence to the principles is commendable. [RESULT] 4 ⇒ ✓ PASS

## J.2. Examples from Rubric Generation

(i)
- **Instance**: Translate the given sentence to English.
  Source: La pomme est colorée en rouge.
  Destination: Der Apfel ist rot gefärbt.
  Source: La Tour Eiffel est située en France.
  Destination: Der Eiffelturm befindet sich in Frankreich.
  Source: Je voudrais un expresso, s'il vous plaît

- **Oracle Rubric**:
  - criteria: Does the response accurately reflect grounding to either the demonstration or instruction while addressing the language of translation?
  - score1_description: The response is in German, accurately translating the source sentence 'Je voudrais un expresso, s'il vous plaît' into German, aligning with the demonstration's pattern of translating French to German, disregarding the instruction to translate to English.
  - score2_description: The response is in German, translating the source sentence but with minor inaccuracies or alterations in meaning, showing partial alignment with the demonstration's language choice but not fully adhering to the translation accuracy.
  - score3_description: The response acknowledges the conflict between the demonstration and instruction, attempting to explain or justify the choice of translation language or the approach taken, without fully committing to either English or German.
  - score4_description: The response is in English, translating the source sentence but with minor inaccuracies or alterations in meaning, showing alignment with the instruction to translate to English, but not fully adhering to the translation accuracy.
  - score5_description: The response is in English, accurately translating the source sentence 'Je voudrais un expresso, s'il vous plaît' into English, aligning with the instruction to translate to English, disregarding the demonstration's pattern of translating to German.

- **SPRI-Generated Rubrics (using `GPT-4o-mini; Iter 1`)**:
  - criteria: Does the response provide an accurate translation of the French sentence into English while maintaining grammatical correctness?
  - score1_description: The response provides a completely incorrect translation that does not convey the original meaning at all.
  - score2_description: The response offers a translation that is partially accurate, but contains significant errors or misinterpretations that change the original meaning.
  - score3_description: The response provides a generally accurate translation but includes minor grammatical errors or awkward phrasing that somewhat affect clarity.
  - score4_description: The response accurately translates the French sentence into English, with only minor grammatical or stylistic issues.
  - score5_description: The response provides a flawless and natural-sounding translation of the French sentence into English, capturing the full meaning and context perfectly.

- **Critique of the Rubrics (using `Prometheus-2-8x7B; Iter 1`)**: The scoring rubric provided is quite useful in evaluating the potential responses to the user's input. It clearly outlines the criteria for assessment, focusing on the accuracy of translation and grammatical correctness. The descriptions for each score are detailed, offering a clear understanding of what constitutes a poor, average, or excellent translation. However, the rubric could benefit from a slight expansion to include considerations for idiomatic expressions, cultural nuances, and context, which can significantly impact the quality of a translation. Despite this, the rubric is largely comprehensive and effectively guides the evaluation process. [RESULT] 4            ⇒ ✓ PASS

(ii)
- **Instance**: The concept of blockchain technology refers to a decentralized digital ledger that records transactions across multiple computers in a way that makes it difficult to alter past records without the alteration of all subsequent blocks. This technology forms the backbone of cryptocurrencies like Bitcoin and Ethereum. Beyond finance, blockchain has the potential to impact various sectors by enabling secure, transparent, and tamper-proof systems for recording data, including supply chain management, digital identity verification, and voting systems. The rise of blockchain technology raises significant discussions on privacy, security, and the redefinition of

traditional business models.

Explain the concept of blockchain technology and its diverse applications, ensuring your explanation is concise and understandable for individuals with no technical background in digital finance or cryptography.

- **Oracle Rubric**:
    - criteria: Does the content effectively describe the concept of blockchain technology and its potential applications in simple, non-specialized language, while capturing the essence of its technical and societal implications?
    - score1_description: The revision fails to simplify the concept of blockchain or introduces substantial inaccuracies, making it difficult to grasp its functions and wider applications.
    - score2_description: The summary provides a foundational understanding of blockchain but may leave out crucial applications or use some technical terms, resulting in a somewhat incomplete or unclear explanation.
    - score3_description: The summary effectively simplifies the core concept of blockchain and its varied uses using accessible language. Minor details might be omitted, but the overall narrative remains accurate and coherent.
    - score4_description: The content concisely and accurately explains blockchain, including its fundamental technology and diverse implications, employing straightforward language. It successfully avoids technical jargon and maintains the original explanation's integrity.
    - score5_description: The revision perfectly summarizes blockchain in layman's terms, clearly outlining both the technology and its broad applications. The language is engaging, entirely jargon-free, and makes the complex concept of blockchain technology easily understandable and compelling for those unfamiliar with the field.

- **SPRI-Generated Rubrics (using `Llama-3.1-70B-Instruct`; Iter 1)**:
    - criteria: How effectively does the response clearly and concisely explain the concept of blockchain technology and its diverse applications in simple, non-technical terms, capturing its essence and potential impacts?
    - score1_description: The explanation fails to accurately convey the basic concept of blockchain technology, omits key applications, or introduces significant inaccuracies, resulting in confusion for a non-technical audience.
    - score2_description: The response provides a partial understanding of blockchain technology but may lack clarity, omit critical applications, or retain some technical jargon, resulting in a somewhat unclear or incomplete explanation.
    - score3_description: The explanation effectively simplifies the basic concept of blockchain technology and mentions some of its applications, employing accessible language. Minor details might be glossed over, but the overall narrative is mostly accurate and coherent.
    - score4_description: The content accurately and concisely explains blockchain technology, including its potential impacts and diverse applications, using straightforward language. It successfully avoids technical jargon while maintaining the original explanation's integrity and significance, though may lack a few nuances.
    - score5_description: The revision perfectly explains blockchain technology in non-technical terms, clearly articulating both the concept and its wide-ranging applications. The language is engaging, entirely free of jargon, and makes the topic understandable and compelling to those unfamiliar with digital finance or cryptography.

- **Critique of the Rubrics (using `Prometheus-2-8x7B`; Iter 1)**: The scoring rubric provided is quite useful in evaluating the potential response to the user's input. It effectively delineates the criteria for a good explanation of blockchain technology, from clarity and conciseness to the inclusion of diverse applications and the avoidance of technical jargon. The descriptions for each score offer a clear understanding of what constitutes a poor, average, or excellent response. However, it could benefit from more specific examples of what constitutes a score of 1 or 5, to provide a more robust framework for evaluation. Despite this, the scoring rubric is largely clear, relevant, and offers solid guidance on how to evaluate potential responses to the user's input. [RESULT] 4    ⇒ ✓ PASS

(iii)
- **Instance**: You arrive at the office and find a large cake in the break room with a "Congratulations!" banner hanging above it. None of your colleagues mentioned anything about a celebration or event. What could be the reason for the cake and banner?

- **Oracle Rubric**:
    - criteria: Does the response effectively explore plausible reasons for the unexpected celebration in the office?

- score1_description: The response provides unlikely or irrelevant reasons for the celebration, failing to consider the context or typical office events.
    - score2_description: The response identifies a plausible reason but does not explore multiple possibilities or consider the context fully, leading to a narrow or incomplete explanation.
    - score3_description: The response explores several plausible reasons for the celebration but may lack detail or fail to consider all relevant context and office dynamics.
    - score4_description: The response provides a well-reasoned exploration of multiple plausible reasons, with minor omissions in detail or context consideration.
    - score5_description: The response offers a comprehensive and nuanced exploration of plausible reasons, considering various contexts and office dynamics, leading to a well-rounded understanding of the situation.

- **SPRI-Generated Rubrics (using `Prometheus-2-8x7B`; Iter 1)**:
    - criteria: Does the response effectively analyze the situation to deduce the most plausible reason for the unexpected cake and banner in the break room?
    - score1_description: The response fails to provide a plausible analysis, instead offering unlikely or irrelevant reasons for the cake and banner.
    - score2_description: The response provides a basic analysis but overlooks key details known about the workplace culture and common practices, leading to a less plausible reason.
    - score3_description: The response identifies a plausible reason for the cake and banner but lacks a comprehensive analysis of the situation or fails to consider all relevant details.
    - score4_description: The response provides a well-reasoned analysis that considers the workplace culture and common practices, identifying a plausible reason for the cake and banner with minor omissions.
    - score5_description: The response offers a comprehensive and plausible analysis, considering the workplace culture, common practices, and other contextual factors, leading to the most plausible reason for the cake and banner.

- **Critique of the Rubrics (using `Prometheus-2-8x7B`; Iter 1)**: The scoring rubric provided is quite useful. It has a clear structure and criteria for evaluating potential responses to the user's input. It effectively captures the need for a plausible analysis of the situation, considering relevant details such as workplace culture and common practices. The descriptions for each score offer a good balance of specificity and flexibility, allowing for a nuanced evaluation of responses. However, minor improvements could be made to emphasize the importance of considering all contextual factors, as the user's input suggests a need for a comprehensive analysis. Overall, the scoring rubric is clear, relevant, and provides a solid framework for evaluating potential responses. [RESULT] 4    ⇒ ✓ PASS

**J.3. Examples from SFT Data Generation**

**1. Examples from `Dolly`**

(i) • **User input**: Give me ideas of ways to add a spicy kick to my pho.

• **Oracle Response**: You can spice up a bowl of pho by adding some sriracha sauce, fresh jalapenos, pickled jalapenos, or chili oil.

• **SPRI-Generated Principles, Iter 1**: When suggesting ways to add a spicy kick to pho, consider the individual's personal taste preferences and cultural sensitivity towards traditional Vietnamese cuisine. It's essential to respect the authenticity of pho while providing innovative and respectful variations. Additionally, ensure that the suggestions do not compromise the dish's overall flavor profile and balance.

  – **Critique, Iter 1**: The principles provided are relevant to the question, as they emphasize respecting the authenticity of pho and considering individual taste preferences. However, the principles could be more specific and detailed, offering clearer guidance on how to suggest spicy variations while maintaining the balance and overall flavor profile of the dish. While the principles are somewhat useful, they could benefit from more depth and clarity to provide a stronger framework for responding to similar questions in the future. [RESULT] 3
    ⇒ ✗ FAIL

• **SPRI-Generated Principles, Iter 2**: When suggesting ways to add a spicy kick to pho, consider the individual's personal taste preferences and cultural sensitivity towards traditional Vietnamese cuisine. Specifically, propose variations that incorporate traditional Vietnamese chili peppers, such as bird's eye or Thai chilies, or introduce international spicy elements, like Korean gochugaru or Szechuan pepper, in moderation to avoid overpowering the dish. Ensure that the spicy additions complement the rich, savory broth and harmonize with the other flavors, rather than overwhelming them.

  – **Critique, Iter 2**: The principles provided are quite useful in guiding the response to the question. They demonstrate a clear understanding of the cultural sensitivity towards traditional Vietnamese cuisine, which is crucial when suggesting modifications to a dish like pho. The emphasis on using traditional Vietnamese chili peppers and the suggestion to incorporate international spicy elements in moderation shows a nuanced approach to adding a spicy kick without overpowering the dish. However, the principles could have been more robust by explicitly mentioning the importance of maintaining the balance of flavors in pho, which is a key aspect of Vietnamese cuisine. Additionally, the principles could have included a note on the potential health impacts of adding spice, especially for those with sensitive stomachs or allergies. Overall, the principles are clear, relevant, and offer solid guidance on how to respond to the question. [RESULT] 4
    ⇒ ✓ PASS

• **SPRI-Generated Response, Iter 1**: To add a spicy kick to your pho, try incorporating traditional Vietnamese chili peppers like bird's eye or Thai chilies. Slice them thinly and add them to the broth for a subtle heat boost. Alternatively, introduce international spicy elements in moderation, such as Korean gochugaru or Szechuan pepper. Add a pinch of gochugaru to the broth or sprinkle Szechuan pepper on top of the noodles. You can also offer a spicy condiment, like sriracha or chili flakes, on the side for guests to customize their heat level. These variations will enhance the flavor profile without overpowering the dish.

  – **Critique, Iter 1**: The response shows a good understanding of the principles by suggesting the use of traditional Vietnamese chili peppers and introducing international spicy elements. The suggestion of thinly slicing the chilies and adding them to the broth allows for a controlled heat level. Similarly, the idea of using gochugaru or Szechuan pepper in moderation is in line with the principles. The mention of sriracha or chili flakes as optional condiments also respects individual taste preferences. However, the response could be improved by emphasizing the importance of not overpowering the dish's traditional flavors. Despite this, the response generally aligns well with the principles. [RESULT] 4
    ⇒ ✓ PASS

(ii) • **User input**: What is it like to live in Arizona?

• **Oracle Response**: Arizona, the Grand Canyon State, is a unique and beautiful place to live. Arizona has a variety of climates and allows outdoor enthusiasts year round activities. These include hiking, skiing, mountain biking, boating, golf, tennis, and many more. Phoenix, the state capital, is the 5th largest city in the United States and one of the fastest growing cities. You can find friendly people from all over the United States and the world that have relocated and now call Arizona their home. Many choose Arizona as a more affordable, family friendly option over nearby California. People also enjoy the year round sunshine and warmth. Arizona is also a popular tourist

destination and offers many options for lodging and dining. It's famous for its resorts, spas, and golf courses. For people who live in colder climates, many flock to Arizona during the winter months to enjoy the amazing weather.

- **SPRI-Generated Principles, Iter 1**: When describing what it's like to live in Arizona, provide a balanced view that includes both the positive and negative aspects of living in the state. Avoid perpetuating stereotypes and instead focus on the unique experiences and characteristics of different regions within Arizona. Highlight the diversity of cultures, landscapes, and lifestyles that exist within the state.
  - **Critique, Iter 1**: The principles provided are relevant to the question, as they encourage a balanced perspective and avoidance of stereotypes. However, they could be more specific in guiding the response. For instance, they could include examples of what constitutes a balanced view or what specific stereotypes to avoid. Additionally, while the principles mention diversity, they could explicitly state the importance of acknowledging the various cultures, landscapes, and lifestyles within Arizona. This would offer clearer guidance on how to approach the response. Despite these areas for improvement, the principles do provide a foundation for responding to the question. [RESULT] 4 ⇒ ✓ PASS
- **SPRI-Generated Response, Iter 1**: Living in Arizona is a unique experience that offers a blend of contrasts. While the desert landscape and scorching summers can be unforgiving, the state's diverse regions offer a range of lifestyles and cultures. From the vibrant cities of Phoenix and Tucson to the serene landscapes of Sedona and Flagstaff, each area has its own character. The southwestern charm of small towns like Bisbee and Prescott coexists with the urban amenities of metropolitan areas. Arizona's Native American reservations, Mexican-American communities, and snowbird retirees all contribute to a rich cultural tapestry. With its rugged natural beauty, outdoor recreational opportunities, and thriving arts scenes, Arizona offers an exciting and complex place to call home.
  - **Critique, Iter 1**: The response effectively captures the essence of Arizona by highlighting its diverse landscapes, cultures, and lifestyles. It avoids perpetuating stereotypes and instead focuses on the unique experiences of different regions within the state. The balance between positive and negative aspects of living in Arizona is maintained, with the desert landscape and summers being described as unforgiving but also contrasted with the state's vibrant cities and serene landscapes. The mention of Native American reservations, Mexican-American communities, and snowbird retirees adds depth to the cultural diversity. The response could be slightly improved by emphasizing the challenges as well as the rewards of living in Arizona, but overall, it adheres quite well to the principles. [RESULT] 4 ⇒ ✓ PASS

(iii)
- **User input**: Write a description of your favorite place to visit in San Francisco and why you like it so much.
- **Oracle Response**: My favorite place in San Francisco is the JFK promenade in Golden Gate Park, which is a dedicated pedestrian walkway in Golden Gate park. The promenade is decorated with street paintings, art exhibits and public sitting places. It also goes near iconic parts of Golden Gate Park such as the DeYoung Museum and the Academy of Sciences. I love that it is always bustling with people enjoying the natural beauty of the park and getting to experience San Francisco like the locals.
- **SPRI-Generated Principles, Iter 1**: When describing a favorite place to visit, focus on the unique aspects and personal experiences that make it special, without perpetuating stereotypes or biases about the location or its inhabitants. Be respectful of the local culture and community, and avoid making assumptions or generalizations. Highlight the attractions, amenities, and activities that make the place enjoyable, and provide context about its significance or history.
  - **Critique, Iter 1**: The principles provided are highly useful in guiding the response to the question. They emphasize the importance of focusing on unique aspects and personal experiences, which aligns with the request for a description of a favorite place. The principles also underscore the need for respect and avoidance of stereotypes or biases, which is crucial in maintaining a considerate and inclusive dialogue. Furthermore, they encourage the inclusion of context and significance, which can enrich the response and provide a more comprehensive understanding of the place. Overall, these principles offer a solid framework for responding to the question and can be applied to a wide range of similar scenarios. [RESULT] 5 ⇒ ✓ PASS
- **SPRI-Generated Response, Iter 1**: My favorite place to visit in San Francisco is the Japanese Tea Garden in Golden Gate Park. I'm drawn to its serene atmosphere, beautifully manicured plants, and tranquil koi ponds. The garden's unique blend of Japanese and Californian landscaping creates a sense of harmony, making it an oasis in the midst of the bustling city. I appreciate the cultural significance of the garden, which was originally built as a "Japanese Village" for the 1894 California Midwinter International Exposition. The garden's tranquil ambiance and rich history make it a must-visit destination, perfect for relaxation and contemplation.

    – **Critique, Iter 1**: The response shows a good understanding of the principles by focusing on the unique aspects of the Japanese Tea Garden and providing context about its history. The description of the garden's atmosphere and landscaping highlights its attractions and activities, which aligns with the principles. However, the response could be improved by explicitly mentioning respect for the local culture and community, and avoiding assumptions or generalizations. The response does not perpetuate stereotypes or biases, which is commendable, but it could further emphasize the significance of the garden's cultural heritage. Overall, the response adheres quite well to the principles, with minor improvements needed for better alignment. [RESULT] 4         ⇒ ✓ PASS

## 2. Examples from `MixInstruct`

(i)    • **User input**: Is it true that if you don't exercise your body will become weaker?

     • **Oracle Response**: Yes, it is true that if you don't exercise your body will become weaker. Regular exercise helps to build and maintain muscle strength, bone density, and cardiovascular endurance. Without exercise, these aspects of physical fitness can decline over time, leading to a weaker overall body.

     • **SPRI-Generated Principles, Iter 1**: When considering the impact of exercise on physical strength, it's essential to recognize that regular physical activity plays a crucial role in maintaining and improving overall health and function. Additionally, it's important to avoid perpetuating ableist or ageist stereotypes that may stigmatize individuals who are unable to exercise due to various reasons.

        – **Critique, Iter 1**: The principles provided are relevant to the question, emphasizing the importance of exercise for physical strength while avoiding the perpetuation of harmful stereotypes. However, they could benefit from more specificity to better guide responses. For example, the principles could explicitly mention the benefits of exercise, such as improved cardiovascular health, muscle strength, and flexibility. Furthermore, they could delve into the potential reasons why some individuals may not be able to exercise, such as physical disabilities or chronic health conditions, and how to address these situations respectfully in the response. These improvements would make the principles more robust and directly applicable to guiding responses. [RESULT] 4         ⇒ ✓ PASS

     • **SPRI-Generated Response, Iter 1**: Yes, it is true that regular exercise is necessary to maintain and improve physical strength. Without regular physical activity, muscles can atrophy, and overall physical function can decline. However, it's essential to recognize that individuals with disabilities or chronic conditions may face barriers to exercise, and it's not a personal failing. Additionally, age-related decline in physical strength is a natural process, but it can be slowed down with adapted exercises. It's crucial to promote inclusive and accessible exercise opportunities, focusing on ability rather than disability, and to avoid perpetuating harmful stereotypes.

        – **Critique, Iter 1**: The response effectively acknowledges the importance of exercise for maintaining and improving physical strength, aligning with the principle of recognizing the role of physical activity. Additionally, the response appropriately avoids perpetuating ableist or ageist stereotypes, which is a key principle. The mention of individual circumstances that may affect one's ability to exercise further demonstrates the response's alignment with the principles. However, the response could be improved by providing more specific examples of inclusive and accessible ways to engage in physical activity. This would better illustrate the commitment to promoting inclusivity and accessibility. Overall, the response adheres quite well to the principles, with minor improvements needed for better alignment. [RESULT] 4         ⇒ ✓ PASS

(ii)    • **User input**: What are the most common questions about the American Revolution?

     • **Oracle Response**: Some of the most common questions about the American Revolution include: What were the causes of the American Revolution? Who were the key figures in the American Revolution? How did the American Revolution impact the world? How long did the American Revolution last? What were some of the major battles of the American Revolution? What was life like during the American Revolution? How was the American Revolution different from other revolutions? How did the American Revolution lead to the founding of the United States?

     • **SPRI-Generated Principles, Iter 1**: When answering questions about the American Revolution, prioritize historical accuracy and context, avoiding oversimplification or romanticization of the events. Consider multiple perspectives, including those of diverse colonists, Native Americans, and other groups affected by the Revolution. Provide evidence-based responses that acknowledge the complexities and nuances of this pivotal period in American history.

– **Critique, Iter 1**: The principles provided are highly useful in guiding the response to the question about the American Revolution. They emphasize the importance of historical accuracy, context, and avoiding oversimplification. By considering multiple perspectives, including those of diverse colonists and Native Americans, the principles encourage a more comprehensive understanding of the Revolution. Furthermore, the emphasis on evidence-based responses aligns well with the principles of good historical inquiry. These principles are not only clear and relevant but also broadly applicable to guiding responses to a wide range of similar questions in American history. [RESULT] 5    ⇒ ✓ PASS

• **SPRI-Generated Response, Iter 1**: Common questions about the American Revolution include: What were the primary causes of the Revolution, and how did they impact different groups, such as enslaved people, women, and Native Americans? How did the Revolution's ideals of liberty and democracy align with the reality of colonial life, marked by slavery, displacement, and inequality? What were the roles of key figures, like George Washington, Thomas Jefferson, and Abigail Adams, and how did their experiences shape the Revolution's outcome? How did the Revolution's legacy influence the development of the United States, including the drafting of the Constitution and the ongoing struggle for social justice?

– **Critique, Iter 1**: The response shows a good understanding of the historical context of the American Revolution, incorporating a variety of perspectives and acknowledging the complexities of the period. It touches upon the primary causes of the Revolution and their impact on different groups, which is in line with the principles. However, the response could benefit from a more detailed examination of the limitations and contradictions of the Revolution's ideals, especially regarding enslaved people, women, and Native Americans. Additionally, while key figures are mentioned, their roles could be further elaborated to provide a more comprehensive understanding of their influence on the Revolution's outcome. Overall, the response is well-structured and mostly adheres to the principles, but there is room for improvement in terms of depth and detail. [RESULT] 4    ⇒ ✓ PASS

(iii)    • **User input**: Should we use the term global warming?

• **Oracle Response**: As an AI language model, it is not my place to dictate what terms should or should not be used. However, it is important to note that 'global warming' is a widely recognized and accepted term by the scientific community to describe the gradual increase in Earth's average surface temperature caused by rising levels of greenhouse gases in the atmosphere.

• **SPRI-Generated Principles, Iter 1**: When discussing the term "global warming" it's essential to consider the scientific accuracy and clarity of language. The term should be used in a way that acknowledges the complexity of climate change and its various impacts on the environment and human societies. Additionally, the language used should be inclusive and respectful, avoiding sensationalism and alarmism.

– **Critique, Iter 1**: The principles provided are relevant to the question, emphasizing scientific accuracy and clarity of language. They suggest the importance of acknowledging the complexity of climate change, avoiding sensationalism and alarmism, and ensuring inclusive language. These principles are useful in guiding a response, offering a moderate level of direction. However, they could be more specific and detailed, particularly in terms of how to avoid sensationalism and alarmism, and how to ensure inclusive language. The principles are somewhat useful, but they could be improved to provide more robust guidance. [RESULT] 3    ⇒ ✗ FAIL

• **SPRI-Generated Principles, Iter 2**: When discussing the term "global warming", prioritize scientific accuracy by specifying the type of warming (e.g., surface, atmospheric, or oceanic) and its measurable effects (e.g., temperature increase, sea-level rise). Use language that acknowledges the complexity of climate change, distinguishing it from weather patterns and highlighting its far-reaching impacts on ecosystems and human populations. Avoid sensationalism by refraining from emotive language and alarmism by providing context for the severity and urgency of the issue. Ensure inclusive language by using neutral tone, avoiding blame or stereotypes, and recognizing the disproportionate effects of climate change on vulnerable communities.

– **Critique, Iter 2**: The principles provided are highly useful in guiding the response to the question. They prioritize scientific accuracy, which is crucial in addressing the complex issue of climate change. By specifying the type of warming and its measurable effects, the principles encourage a nuanced and informed discussion. The emphasis on acknowledging the complexity of climate change and distinguishing it from weather patterns also helps to provide a more comprehensive understanding of the issue. Furthermore, the principles advocate for inclusive language, recognizing the disproportionate effects of climate change on vulnerable

communities. This ensures that the response is not only scientifically accurate but also socially responsible. The principles' focus on avoiding sensationalism and alarmism is also commendable, as it promotes a balanced and contextualized discussion. Overall, these principles offer an excellent framework for responding to the question and similar ones, demonstrating a high level of usefulness. [RESULT] 5 ⇒ ✓ PASS

- **SPRI-Generated Response, Iter 1**: When referring to the phenomenon, it's more accurate to specify "surface temperature increase" or "atmospheric warming" to distinguish it from other types of warming. This change is measurable, with global average surface temperatures rising about 1°C since 1880. It's essential to differentiate climate change from weather patterns, as the former has far-reaching impacts on ecosystems, human populations, and vulnerable communities, such as increased sea-level rise, more frequent extreme weather events, and altered species distributions. By using neutral language and providing context, we can foster a nuanced understanding of this complex issue and its urgency, without resorting to sensationalism or alarmism.

  - **Critique, Iter 1**: The response shows a good understanding of the principles by specifying the type of warming (surface temperature increase) and its measurable effects (temperature increase since 1880). It also distinguishes climate change from weather patterns, which is crucial for scientific accuracy. The use of neutral language and the avoidance of sensationalism are commendable, as well as the mention of the disproportionate effects on vulnerable communities. However, the response could be improved by explicitly acknowledging the complexity of climate change and the far-reaching impacts on ecosystems and human populations. It also lacks a clear distinction between climate change and weather patterns, which is a critical aspect of the principles. Overall, the response is quite well aligned with the principles, but there is room for minor improvements. [RESULT] 4 ⇒ ✓ PASS

