# OpenReview forum: "SPRI: Aligning Large Language Models with Context-Situated Principles"
_ICML.cc/2025/Conference — ICML 2025 poster_

### Official Review · Reviewer_HZMY · 2025-03-11

**Overall Recommendation:** 3

**Summary:**

Large Language Models (LLMs) often require guiding principles to ensure their responses are well-aligned and contextually appropriate. While prior work has leveraged predefined principles or constitutions for synthetic data generation, these approaches often fail to adapt to situation-specific needs.
SPRI is a framework that dynamically generates query-specific constitutional principles to guide LLM responses dynamically. Unlike static principle-based approaches, SPRI adapts to context for better alignment.

Key Findings:
-Models using SPRI perform as well as those using expert-crafted principles.
-SPRI matches human evaluation rubrics and compares favorably with LLM-judge methods.
-SPRI-generated synthetic data enhances LLM performance on TruthfulQA.

**Claims And Evidence:**

yes they are (experiments illustrate well the claims)

**Essential References Not Discussed:**

No

**Experimental Designs Or Analyses:**

yes

**Methods And Evaluation Criteria:**

yes they do

**Other Comments Or Suggestions:**

I also have a request for concrete examples illustrating the SPRI process. Apprendix I shows generated principles/responses but not at each step of the staged approach. Would be nice to show generated principles along the two-step refinement process in Stage 1 (1) initial principles generated and (2) refined principles as described in Section 3. Then, in Stage 2, show (3) a principle-constrained response and (4) its refinement after applying the critic. This would clarify how SPRI iteratively improves both principles and responses.


typos or remarks:
- l146 pertaining  => pretraining
-tab 2: you should mention in caption that Pearson’s correlation coefficient is used
-tab 4: i suppose amount of fine-tuning data is different for each line (for instance oracle response vs SPRI), would be nice to provide this information in the table or the caption.

**Other Strengths And Weaknesses:**

One of my main concern is whether SPRI’s fine-grained, user-input-level principles are sustainable. Unlike static constitutions, the dynamic generation of query-dependent principles raises questions about inference time (cf staged approach described in section 3). I’m also wondering whether principles should be retained or discarded after use (nothing is said about this). If kept, managing conflicts with pre-existing principles becomes an issue.
In other words I’m wondering whether query-level granularity is the right approach or if it’s too fine-grained to be practical; a discussion on this would help clarify this.


I’m also wondering whether the assumed scenario—starting from zero principles available—is the most realistic one. In practice, isn’t it more common to begin with a predefined set of expert principles and then refine or adapt them to the specific situation rather than generating them from nothing? Would SPRI benefit from integrating existing expert knowledge as a foundation rather than recreating principles entirely (i have the feeling Appendix C describes such a case where Default Seed Principles are presented)?

**Questions For Authors:**

is  Prometheus-2-8x7B (Kim et al., 2024b)  the best choice for the critic model ? how about gpt4o-mini, claude or llama model ? What motivated this choice ?

**Relation To Broader Scientific Literature:**

Related work section is fine

**Theoretical Claims:**

not many theoretical claims (though of formalization of the method and pseudo-code are provided)

---

> ### Author Rebuttal · Authors · 2025-04-01
>
> Thank you for your positive comments on the innovation of SPRI, which dynamically generates context-adaptive principles to align LLMs while relying on minimal-to-no human supervision. We are also grateful for your acknowledgment of the experimental results — a strength that other reviewers also appreciated.
>
> > ### 💡**Inference Cost of SPRI:**
>
> Please refer to the rebuttal to Reviewer dFiR for an in-depth discussion of the inference cost of SPRI.
>
> > ### 💡**Are Principles Retained or Discarded after Use?**
>
> SPRI *discards* principles that are not satisfactory, but we note that they are the stepping stones to the final satisfactory principles. To be more specific, as Figure 2 shows, each set of principles generated in Stage 1 of SPRI is first scrutinized by the critic model. If the critic model deems the principles not useful enough to guide the response to the query, we ask the base model to refine these principles based on the critic model’s feedback. The old principles are then discarded, but they also serve as the basis for the base model’s refinement and, subsequently, the final principles. Nevertheless, if the critic model deems the principles satisfactory, they are kept and used as guidance for responding in Stage 2. We will also better illustrate this process in Appendix A Algorithm 1.
>
> > ### 💡**Query-Level Granularity:**
>
> As shown in Figure 1, when Reappraising for a person in distress, generic rules don’t apply to the context at all, whereas expert-crafted prompts demand human expertise and are time-consuming to write. Similarly, in BiGGen Bench, in order to come up with query-specific evaluation rubrics to improve the performance of LLM judges, Kim et al. (2024) needed to hand-craft instances with at least 28 annotators. In comparison, SPRI approaches the performance of these expert-guided methods and outperforms static-rule-based ones. In terms of cost, SPRI is slightly more expensive than the static ones, but a lot cheaper and more practical than employing annotators.
>
> > ### 💡**Is Starting from Zero Principles the Realistic Approach?**
>
> While SPRI is *not* given any expert principles as the starting point, we kindly point out that we include seed examples in the initial principle-generation process of SPRI. For Sec 4.1 Cognitive Reappraisal, a single oracle reappraisal constitution was provided as the seed example (line 202); whereas for Sec 4.2 Fine-Grained Rubrics, 3 instance-rubric pairs from BiGGen Bench were used as seed examples (line 271).
>
> As a matter of fact, SPRI does benefit from having access to existing expert knowledge, but we would point out that SPRI still achieves comparable performance even without it. As shown in Table 3 of Sec 4.3 where we conducted ablation studies on the effects of the seed examples in tasks that require complex guidance, removing seed examples entirely leads to an average performance degradation of 4.13% in alignment for reappraisals and 13.37% in Pearson’s correlation for rubric generation. *This demonstrates that SPRI can still achieve comparable performance even without any human supervision.* Similarly, substituting the default principles (shown in Appendix C) as seed examples leads to an average performance decrease of 4.01% in alignment and 12.35% in Pearson’s correlation for rubric generation. These results highlight the robustness of SPRI, as the default principles are not relevant to these tasks at all — in fact, they can be seen as distractions to SPRI’s principle generation.
>
> > ### 💡**More Concrete Examples Illustrating the Critique-Refinement Process of SPRI:**
>
> While Appendix I exhibits the generated principles & responses from SPRI for each of the 3 tasks that we experimented on, we agree that more concrete examples — involving the principles/responses generated at each step of the 2 stages in SPRI — would better illustrate how SPRI iteratively improves both its principles and responses. Although we cannot attach an example of the full cycle of the critique and refinement of the principles & responses due to the character limitations in the rebuttal, we will make sure to further incorporate them in Appendix I of the camera-ready paper. Thank you for the suggestion!
>
> > ### 💡**Why We Chose Prometheus-2-8x7B as the Critic Model:**
>
> We select Prometheus-2-8x7B as the critic model in our experiments because it is a good-performing model specifically trained to be an LLM-judge ([Kim et al., EMNLP 2024](https://aclanthology.org/2024.emnlp-main.248.pdf)). Besides, the MoE nature of this model makes it relatively light-weight, yielding faster run-time during critiquing. However, we agree that other models, such as GPT-4o-mini, could be used as an alternative critic model for SPRI. In fact, as we showed in the tables in the rebuttal to Reviewer dFiR, the computational cost of SPRI can also be brought down a lot if we were to choose a cheap yet powerful model (like GPT-4o-mini). We leave this interesting question to future work.

---

> > ### Comment · Reviewer_HZMY · 2025-04-07
> >
> > Thanks for your responses to my comments, and for the clarifications especially related to inference cost of SPRI.
> >
> > However I think there may have been a misunderstanding around one of my questions. Specifically, when I wrote: "I’m also wondering whether principles should be retained or discarded after use (nothing is said about this). If kept, managing conflicts with pre-existing principles becomes an issue..."
> >
> > What I meant was: once a query has been answered, do you log or retain the principles that were finally used (which might be useful for future queries maybe)? Or do you always start from a new, empty set of principles for each new query? Based on your previous reply, I understand that you did not consider building such a memory of principles, which is fine, but I’d still be curious to hear your thoughts about this...

---

> > > ### Author Response · Authors · 2025-04-08
> > >
> > > Thank you for clarifying the question! You are correct — we don’t log or retain the principles for future queries. The reason why we start anew for each query is that the generated principles from SPRI are *specific* to each query (as you saw in Appendix I) — and this is exactly what SPRI is designed to do. This specificity of the principles proves to be beneficial for tasks like Reappraisal and Instance-Specific Evaluation, where our method outperforms methods that rely on generic static rules. Nevertheless, we agree that it is interesting to explore how we can reuse the generated principles as the starting point for new queries. But the trick here is to determine the threshold of generalizability and specificity in the principles.

---

### Official Review · Reviewer_78Ye · 2025-03-14

**Overall Recommendation:** 2

**Summary:**

The proposed SPRI framework automates real-time generation of context-specific guiding principles for LLM alignment, minimizing reliance on human expertise while addressing the limitations of generic predefined rules. SPRI achieves performance on par with expert-crafted principles in domain-specific tasks.

**Claims And Evidence:**

The paper’s core idea is both novel and important – automating alignment guidance per query is a clear step forward for making LLMs safer and more reliable without constant human supervision. The authors also articulate this contribution well, contrasting it with static-rule methods and highlighting SPRI’s adaptability​

**Essential References Not Discussed:**

Weaknesses: There is little to fault in the related work coverage. One minor point is that the paper could have explicitly cited or discussed the concept of “alignment tax” earlier, since it is later mentioned when discussing results​. Works like Askell et al. (2021) are cited in passing​
, but a brief explanation that aligned models can sometimes perform worse on certain benchmarks (the alignment tax phenomenon) would give even more context to why maintaining performance on broad tasks (as SPRI does) is important. However, this is a very subtle critique and does not detract from the overall quality of the related work section. Another possible addition could be a reference to the emerging idea of using multiple models or modules for self-checking (somewhat akin to “debate” or multi-agent alignment techniques), but those are less directly relevant to context-situated principles and their omission is understandable. In summary, the paper adequately reviews prior research and positions itself clearly. It builds directly on known limitations of past methods and cites those sources, ensuring the reader recognizes SPRI’s place as a next-step in alignment research.

**Experimental Designs Or Analyses:**

The paper doesn’t report the runtime or cost, so in real deployment this overhead could be non-trivial. If principles are generated anew each time, an aligned response might take, say, 2–5x the compute of a normal response. This trade-off isn’t discussed in the results. However, given the significant gains in alignment and quality, the extra cost might be justified for critical applications. In summary, the results section is a clear strength of the paper – it provides compelling evidence that SPRI is effective across different challenging alignment tasks, with only minor questions left about evaluation depth and runtime performance.

**Methods And Evaluation Criteria:**

One possible critique of the contributions is that the approach’s complexity (using a critic model and iterative refinement) might make deployment non-trivial – the paper does not deeply discuss the computational cost or latency of generating principles for each query. In practice, generating multiple critique loops per query could be expensive, which might limit real-world significance unless the benefits clearly outweigh the cost. Additionally, while SPRI is novel, it does combine existing ideas (e.g. using an AI feedback loop similar to RLAIF or self-refinement). The true innovation is in what is being refined (principles), but some may view the method as an incremental engineering of known techniques. Nevertheless, the paper makes a strong case that this incremental combination yields qualitatively new capabilities in alignment.

**Other Comments Or Suggestions:**

Refer to the weakness.

**Other Strengths And Weaknesses:**

Weakness: On the weaker side, the paper could discuss practical considerations more, such as the computational cost of SPRI’s iterative process or how it might scale to real-world deployment with many users. Additionally, while the evaluations were mostly automatic for feasibility, a bit more human evaluation (even if anecdotal or case-study based) could further strengthen confidence in the quality of SPRI-guided outputs (especially in sensitive tasks like counseling). These weaknesses are relatively minor and can be addressed in future work.

**Questions For Authors:**

Refer to the weakness.

**Relation To Broader Scientific Literature:**

The topic of the paper is important.

**Theoretical Claims:**

No Theoretical Claims in this paper.

---

> ### Author Rebuttal · Authors · 2025-04-01
>
> We are grateful for your valuable feedback! We appreciate your recognition of the novelty and importance of SPRI in automating the alignment guidance per query to enhance the safety and reliability of LLMs with minimal human supervision, which other reviewers concurred with. Thank you also for pointing out the strength of the results section, which clearly demonstrates that SPRI significantly outperforms static-rule-based methods and achieves performance on par with approaches employing oracle principles, yielding qualitatively new capabilities in alignment. We address your comments below.
>
> > ### 💡**Computational Cost of SPRI:**
>
> Due to space limit, please refer to the rebuttal to Reviewer dFiR for an in-depth discussion of the token usage & computational cost of SPRI.
>
> We would like to additionally highlight to reviewers that *SPRI reduces the heavy dependence on human supervision and, therefore, significantly lowers the costs in both time and money*. For example, having clinical psychologists write prompts for Cognitive Reappraisal (section 4.1) and crowd-sourcing fine-grained evaluation rubrics for BiGGen Bench (section 4.2) would be considerably more costly — exceeding the cost of SPRI by a great extent. This is precisely what SPRI is designed to automate, and results show that it can achieve comparable performance even with the minimal amount of human guidance involved.
>
> > ### 💡**More Human Evaluation:**
>
> We enforce automatic evaluations as it would be more feasible and easier to scale up. Nevertheless, we kindly point out to reviewers that *for the task of Instance-Specific Rubrics (Section 4.2), we carried out the evaluation based on Pearson’s correlation against gold human ratings (see line 317)*. Specifically, the LM-judges’ scores for a total of 2,780 BiGGen Bench examples — judging either using instance-specific rubrics generated by SPRI, or other static-rubric approaches — are compared against the human ground truth labels from these BiGGen Bench examples. Results suggest that SPRI outperforms all instance-agnostic static rubrics across all base models we tested. In addition, SPRI correlates highly with human gold truth ratings on these 2,780 BiGGen Bench examples, with statistical significance on almost all capabilities across all the base models (see Appendix Table 6).
>
> Besides, while for the task of Cognitive Reappraisal (Section 4.1), the evaluation was carried out on relatively fewer data, GPT-4-0613 has been shown to correlate highly with expert humans on the evaluation criteria ([Zhan et al., COLM 2024](https://openreview.net/forum?id=yK8MT91dQY)). On the other hand, human evaluation of these Reappraisal responses would require annotators with strong expertise in clinical psychology, and it is both time-consuming and costly for them to evaluate all the responses we gathered using various methods from the 4 models we tested. The Cognitive Reappraisal results also show similar trends as fine-grained Instance-Specific Rubrics, where SPRI consistently outperforms methods without access to oracle guidance.
>
> In addition, we also kindly refer reviewers to Appendix I of the paper, which shows examples of the generated principles & responses from SPRI for each of the 3 tasks that we conducted in the paper. As Reviewer HZMY noted, these examples paint an intuitive picture of the ability of SPRI to adapt to context, as well as the quality of its responses compared side by side with the oracle ones. In the camera-ready version of the paper, we will further include all the critique-refinement responses from SPRI, from both the principle- and response-generation stages.
>
> > ### 💡**Essential References to Discuss:**
>
> Thank you for appreciating our positioning of SPRI with respect to prior literature! In the camera-ready version, we will mention “alignment tax” earlier in the paper, and expand our discussion on [Askell et al. (2021)](https://arxiv.org/abs/2112.00861). We agree that it would provide readers with a clearer idea as to why it is significant that SPRI can enhance performance on TruthfulQA while preserving results on other benchmarks when fine-tuning LLMs. Additionally, we will also include a more in-depth discussion on related literature such as self-checking.

---

### Official Review · Reviewer_dFiR · 2025-03-14

**Overall Recommendation:** 3

**Summary:**

This paper proposes a novel framework named SPRI for aligning LLMs with human preferences. The framework operates through a two-stage collaborative process between models:
1.  A base model dynamically generates context-specific principles tailored to each input query, iteratively refined through feedback from a critic model.
2. The finalized principles are then utilized to modify the base model’s responses, ensuring alignment.
Compared to alignment methods requiring extensive training or predefined rules, SPRI offers a intuitive solution. The authors validate SPRI’s effectiveness through three key experiments:
1. SPRI-derived principles achieve parity with expert-crafted guidelines in complex tasks, demonstrating its capability to generate context-aware guidance.
2. SPRI-generated evaluation rubrics correlate strongly with human-annotated criteria, outperforming prior LLM-as-a-judge frameworks in granularity and contextual relevance.
3. Fine-tuning LLMs on SPRI-generated synthetic data yields significant improvements in truthfulness metrics (e.g., TruthfulQA) while maintaining performance on general benchmarks, showcasing its potential for scalable alignment.

**Claims And Evidence:**

The majority of the claims presented in this paper are well-supported by detailed experimental evidence, demonstrating the robustness of the proposed approach.

**Essential References Not Discussed:**

N/A

**Experimental Designs Or Analyses:**

I would encourage the authors to include more experiments where human evaluators, rather than LLMs, serve as judges. This would provide stronger credibility and make the findings more persuasive.

**Methods And Evaluation Criteria:**

Increasing the amount of data in Section 4.1 (beyond the current 30 instances) would enhance the soundness of the results.

**Other Comments Or Suggestions:**

1. Since SPRI naturally increases the length of the context, I suggest that the authors provide a more detailed discussion of the associated computational costs and potential trade-offs.
2. In Section 4.1, the authors evaluate SPRI’s performance on a cognitive reappraisal task using a relatively small dataset. I encourage the authors to include additional experiments demonstrating SPRI’s effectiveness on more complex tasks with larger datasets.

**Other Strengths And Weaknesses:**

1. The collaborative framework between LLMs significantly increases the number of additional tokens during interaction, reducing the effective context window available to users and increasing computational costs.
2. The experiment in Section 4.1 is conducted on a relatively small dataset of only 30 instances, which may undermine the reliability and robustness of the results. Expanding the dataset would strengthen the validity of the findings.

**Questions For Authors:**

1. I would like to know whether the authors have conducted experiments where human evaluators directly assess the results instead of relying on the evaluation schema from Zhan et al. (2024). Although this evaluation schema has been shown to have a high correlation with human judgments, I would still prefer to see results obtained from actual human evaluations for further validation.
2. The authors state that the critic model in SPRI can be a smaller-scale model, but the paper does not provide a detailed discussion on this aspect. I am interested in understanding how the choice of critic models with different parameter sizes affects the overall performance of the framework.

**Relation To Broader Scientific Literature:**

This paper makes a meaningful contribution to the problem of aligning LLMs, particularly in the direction of generating context-adaptive principles. The authors discuss existing approaches in the field in the Related Work section and highlight the advantages of SPRI over these methods. However, the comparison with other approaches could be further improved. For instance, a more detailed comparison between SPRI and other LLM alignment methods, particularly those that induce actual changes in model weights, would strengthen the analysis. Additionally, a more in-depth discussion of SPRI’s positioning within the broader paradigm of self-aligned LLMs would provide readers with a clearer understanding of its contributions.

**Theoretical Claims:**

N/A

---

> ### Author Rebuttal · Authors · 2025-04-01
>
> Thank you for your appreciation of the meaningful contribution SPRI makes toward aligning LLMs with context-situated principles while relying on little-to-no human effort. We are also grateful for your recognition of SPRI’s robustness, which is supported by detailed experimental results — a key strength of the paper, as other reviewers also recognized.
>
> > ### 💡**Number of Tokens & Computational Costs Induced by SPRI:**
>
> In Appendix Tables 5 & 6, we reported SPRI’s average model calls for Cognitive Reappraisal and Instance-Specific Rubric Evaluation. However, we agree that it is important to discuss the token usage and computational costs of SPRI in more depth. Therefore, we provide a comparison table of SPRI vs other methods for each task, which will be included in the camera-ready version. We report the average model calls & input/output token usage per response for the base and critic models, as well as the estimated total cost to carry out an entire task. We estimate the cost using OpenAI’s API pricing for GPT and TogetherAI’s pricing for open-source models.
>
> 1. **Cognitive Reappraisal:** *(base model = GPT-4o-mini, critic model = Prometheus-2-8x7B)*
> ||Model Calls|Input Tokens (Base Model)|Output Tokens (Base Model)|Base Model Total Cost|Input Tokens (Critic Model)|Output Tokens (Critic Model)|Critic Model Total Cost|
> |:-|:-:|:-:|:-:|:-:|:-:|:-:|:-:|
> |vanilla|1|299|94|$0.003|--|--|--|
> |self-refine|6|2,106|465|$0.018|--|--|--|
> |oracle|6|4,280|1,421|$0.045|--|--|--|
> |SPRI|4.5|639|220|$0.007|1,537|281|$0.033|
>
> 2. **Instance-Specific Rubric Evaluation:** *(base model = GPT-4o-mini, critic model = Prometheus-2-8x7B)*
> ||Model Calls|Input Tokens (Base Model)|Output Tokens (Base Model)|Base Model Total Cost|Input Tokens (Critic Model)|Output Tokens (Critic Model)|Critic Model Total Cost|
> |:-|:-:|:-:|:-:|:-:|:-:|:-:|:-:|
> |vanilla|1|568|99|$0.403|--|--|--|
> |self-refine|6|4,147|619|$2.762|--|--|--|
> |MT-Bench rubric|1|469|200|$0.530|--|--|--|
> |FLASK rubric|1|636|103|$0.437|--|--|--|
> |oracle|1|707|105|$0.469|--|--|--|
> |SPRI|4.9|1,720|317|$1.247|2,642|282|$4.877|
>
> 3. **SFT:** *(base model = Llama-3-70B-Instruct, critic model = Prometheus-2-8x7B; the estimate is based on using Dolly as the starting instruction-tuning dataset)*
> ||Model Calls|Input Tokens (Base Model)|Output Tokens (Base Model)|Base Model Total Cost|Input Tokens (Critic Model)|Output Tokens (Critic Model)|Critic Model Total Cost|
> |:-|:-:|:-:|:-:|:-:|:-:|:-:|:-:|
> |direct response|1|113|61|$1.9|--|--|--|
> |self-instruct|1|1,046|99|$12.2|--|--|--|
> |self-align|1|1,116|143|$13.4|--|--|--|
> |self-refine|2.1|139|69|$2.2|--|--|--|
> |SPRI|5.0|1,077|167|$13.3|1,363|258|$11.5|
>
> Compared to self-refine, SPRI incurs fewer model calls in tasks that demand complex principles, whilst maintaining significantly stronger performance (see paper tables 1 & 2). Specifically, in **(1) Cognitive Reappraisal**, the base model’s token usage under SPRI is considerably less than those employing oracle principles, and the total cost for the base model is the second cheapest to vanilla prompting. For **(2) Instance-Specific Rubric Evaluation**, while the base model’s cost for SPRI is higher than other context-agnostic approaches, the average number of model calls of SPRI is still less than self-refine. For **(3) SFT**, we can see that the input/output token usage of SPRI is similar to that using self-instruct and self-align, and the total cost is comparable too.
>
> We observe that the additional cost SPRI incurs mainly comes from the critic model, but this can be mitigated by using a cheaper critic model. We chose Prometheus-2-8x7B because it was specifically trained for LLM-judge. However, the critic model in SPRI can also be a smaller-scale model, such as GPT-4o-mini, and this would significantly reduce the cost of SPRI. We leave the interesting question of the tradeoff between the size of the critic model and the performance of SPRI to future work. As Reviewer 78Ye suggested, given the significant gains in alignment and quality of SPRI, the extra cost can be justified for critical applications.
>
> > ### 💡**Human Evaluation and Amount of Eval Data for Reappraisal:**
>
> We did not conduct human evaluation for Reappraisal due to the need for psychological expertise among evaluators and the time-consuming nature of the task. Nonetheless, GPT-4 has been shown to correlate highly with human experts on these 30 evaluation data. Moreover, we highlight that for Instance-Specific Rubrics, we carried out the evaluation for ~2.8k examples based on Pearson’s correlation against gold human ratings. Please refer to the rebuttal to Reviewer 78Ye for a more detailed discussion of the human evaluation.
>
> > ### 💡**Comparison to Other Approaches:**
>
> SPRI differs from alignment methods that require updates in model weights in that it doesn’t require parameter updates, which makes it more efficient at test time. We will also add a more detailed discussion of SPRI’s positioning in the self-aligned paradigm.

---

> > ### Comment · Reviewer_dFiR · 2025-04-02
> >
> > Thank you for the additional results, IMHO a comprehensive human evaluation involving expert efforts is still necessary for a proper meta-evaluation. I think my current ratings already accurately reflect my judgement on this work.

---

> > > ### Author Response · Authors · 2025-04-08
> > >
> > > Thank you so much for your review again!

---

### Decision · Program_Chairs · 2025-05-01

**Decision:**

Accept (poster)

**Comment:**

I liked this paper. I think the topic of LLM alignment is of critical importance, and this paper contributes novel ideas and tidy experiments. The framework presented here probably isn't the long-term solution to alignment, but I can imagine several ways to improve, expand and refine this work -- and that gives me hope that the work will have positive impact.

The paper does have weaknesses, as identified by the reviewers and as reflected in their scores. However, I think the strengths just barely outweigh the weaknesses. If accepted, I strongly encourage the authors to respond to those concerns.

To be clear, I have my own concerns about this paper, which were not raised by the reviewers (for example, it seems like the "meta prompting" needed becomes a new set of static principles), but overall, I think that the contributions are interesting enough to disseminate to the research community.